



# Characterizing nonlinear, nonstationary, and heterogeneous hydrologic behavior using Ensemble Rainfall-Runoff Analysis (ERRA): proof of concept

James W. Kirchner[1,2,3]

[1]Dept. of Environmental Systems Science, ETH Zurich, CH-8092 Zürich, Switzerland
[2]Swiss Federal Research Institute WSL, CH-8903 Birmensdorf, Switzerland
[3]Dept. of Earth and Planetary Science, University of California, Berkeley, CA, 94720, U.S.A.

*Correspondence to*: James W. Kirchner (kirchner@ethz.ch)

**Abstract.** A classical approach to understanding hydrological behavior is the unit hydrograph and its many variants, but these often assume linearity (runoff response is proportional to effective precipitation), stationarity (runoff response to a given unit of rainfall is identical, regardless of when it falls), and spatial homogeneity (runoff response depends only on spatially averaged precipitation). In the real world, by contrast, runoff response is typically nonlinear, nonstationary, and spatially heterogeneous. Quantifying this nonlinearity, nonstationarity, and spatial heterogeneity is essential to unraveling the mechanisms and subsurface properties controlling hydrological behavior.

Here I present proof-of-concept demonstrations illustrating how nonlinear, nonstationary, and spatially heterogeneous rainfall-runoff behavior can be quantified, directly from data, using Ensemble Rainfall-Runoff Analysis (ERRA), a data-driven, nonparametric, model-independent method for quantifying rainfall-runoff relationships across a spectrum of time lags. I show how ERRA uses nonlinear deconvolution to quantify how catchments' runoff response varies with precipitation intensity, and also to estimate their precipitation-weighted runoff response distributions. I further illustrate how ERRA combines nonlinear deconvolution with demixing techniques to reveal how runoff response depends jointly on precipitation intensity and nonstationary ambient conditions, including antecedent wetness and vapor pressure deficit. I demonstrate how ERRA's demixing techniques can be used to quantify spatially heterogeneous runoff response in different parts of a catchment, even if those subcatchments are not separately gauged. I also illustrate how ERRA's broken-stick deconvolution capabilities can be used to quantify multiscale runoff responses that combine hydrograph peaks lasting for hours and recessions lasting for weeks, well beyond the average spacing between storms.

ERRA can unscramble these multiple effects on runoff response even if they are overprinted on each other through time, and even if they are corrupted by autoregressive moving average (ARMA) noise. Results from this approach may be informative for catchment characterization, process understanding, and model-data comparisons; they may also lead to a better



understanding of storage dynamics and landscape-scale connectivity. An R script is provided to perform the necessary calculations, including uncertainty analysis.

# 1 Introduction

## 1.1 Response time versus transit time

When a substantial rainstorm hits a catchment, streamflow will often rise, peak, and recede within a matter of hours or days. However, much of the rainwater from that storm will remain within the catchment for months or even years, affecting the composition of streamflow long after the storm's effects on discharge rates have faded away. Therefore streamflow often primarily _consists of_ precipitation that fell long ago, but is primarily _mobilized by_ rain that fell much more recently. Thus

timescales of hydrologic response are often much shorter than timescales of transport. This decoupling of timescales has been recognized since at least the 1960's (e.g., Brown, 1961; Martinec, 1975; Rodhe, 1981), but understanding its underlying mechanisms remains a central challenge in catchment hydrology (Kirchner, 2003; McDonnell and Beven, 2014).

The contrast between hydrologic response and transport, and between their respective timescales, can be simply illustrated

through the behavior of a simple conceptual catchment model. Figure 1 shows how a simple nonlinear two-box model responds to a hypothetical sequence of 10 identical rainstorms (Kirchner et al., 2023). The underlying model is not intended to realistically mimic any particular catchment; instead it provides a very simple representation of catchment heterogeneity and nonlinearity (which are ubiquitous in real-world settings), and allows their consequences to be traced exactly. The model used to generate Fig. 1 consists of two identical nonlinear reservoirs connected in series. In the interests of simplicity,

evapotranspiration is excluded, and all precipitation enters the upper box. Half of the drainage from the upper box bypasses the lower box and flows directly to the outlet; the other half recharges the lower box, which in turn drains to the outlet.

The left-hand side of Fig. 1 shows how each rainstorm affects future discharge rates, whereas the right-hand side shows how much water from each rainstorm is present in future streamflow. Four individual storms are highlighted in contrasting

colors. The effects of each event on future runoff volumes are quantified by effect tracking – that is, by running the model with and without each event, and comparing the resulting hydrographs. The differences between the hydrographs with and without the four highlighted rainstorms, quantifying the stream's runoff response to them, are shown by the matching colored bands in Fig. 1a. The runoff responses to all 10 storms are shown in Fig. 1c, each expressed relative to a baseline of zero so that their magnitudes can be more easily compared. Each of these runoff responses has the same volume; since the model

ignores evapotranspiration, each millimeter of precipitation must eventually become a millimeter of additional streamflow.



One can see that owing to the nonlinearities in the underlying model, the runoff responses that are initially larger decay away faster. One also can see that the initial runoff responses are larger for storms that fall when the model catchment is already wet, and thus discharge is already high; because the model is nonlinear, runoff response to a given storm depends on how

large, and how recent, previous storms were. The model's nonlinearity further implies that runoff response also depends on future precipitation, as one can see from Fig. 1c. In one particularly clear example, the largest runoff response to storm #8, shown in red, does not come at the end of storm #8 but instead comes during storm #9, illustrating how storm #9 amplifies storm #8's effects. When these runoff response curves are synchronized according to the times that the precipitation fell and are normalized by the precipitation rate (Fig. 1e), they can be considered as *runoff response distributions* (RRDs), which

quantify how the runoff response to a unit of precipitation is distributed over future time. In nonlinear systems, these runoff response distributions will be time-varying, because they depend on the size and timing of both preceding and subsequent precipitation inputs, as shown in Fig. 1e. Nonetheless, one may summarize the time-varying RRDs for a given ensemble of events by averaging them together, resulting in the *ensemble runoff response distribution* shown in dark blue in Fig. 1e.

In contrast to the short-lived runoff response shown in the left-hand side of Fig. 1, the right-hand side shows that the same storms have much more persistent effects on the composition of streamflow itself. That is, much of the water that is added to the catchment by each storm remains stored in the landscape, and continues to be exported via runoff, long after the storm's effects on runoff water fluxes have become unmeasurably small (compare Figs. 1a and 1b). The colored bands in Figs. 1b and 1d show how much of the model's runoff is composed of water that entered the catchment during each storm (or before

all of them, as indicated in Fig. 1b by light gray). As Fig. 1b shows, the gray "old" water makes up a significant fraction of streamflow throughout all 10 storm events, even though the combined volume of all 10 storms is more than twice the model's total storage volume. "Old" water persists longer than one might expect, because older and newer waters are continually mixed (thus some older water always remains), and half of the upper box's output bypasses the lower box entirely, so the lower box flushes slower than the catchment as a whole. Figure 1d shows the contributions of all 10 storms

to future streamflow, relative to a baseline of zero so that they can be more easily compared (analogous to Fig. 1c). The water contributed by each storm persists in runoff throughout the whole sequence of storms, with each storm mobilizing the water contributed by its predecessors. Storms that fall when the catchment is wetter generate sharper peaks in Fig. 1d, and also mobilize more water from prior storms, compared to storms that fall during drier conditions. Thus each storm's distribution of transit times (between when water enters the catchment and subsequently leaves it) depends not only on the

size and timing of previous storms (which determine the antecedent wetness of the catchment), but also the size and timing of subsequent storms (which determine when, how much, and how quickly the water remaining in the catchment will be mobilized). Thus these *transit time distributions* (TTDs) are time-varying, but they can nonetheless be summarized by averaging them together for an ensemble of events, resulting in the *ensemble transit time distribution* shown in dark blue in Fig. 1f.




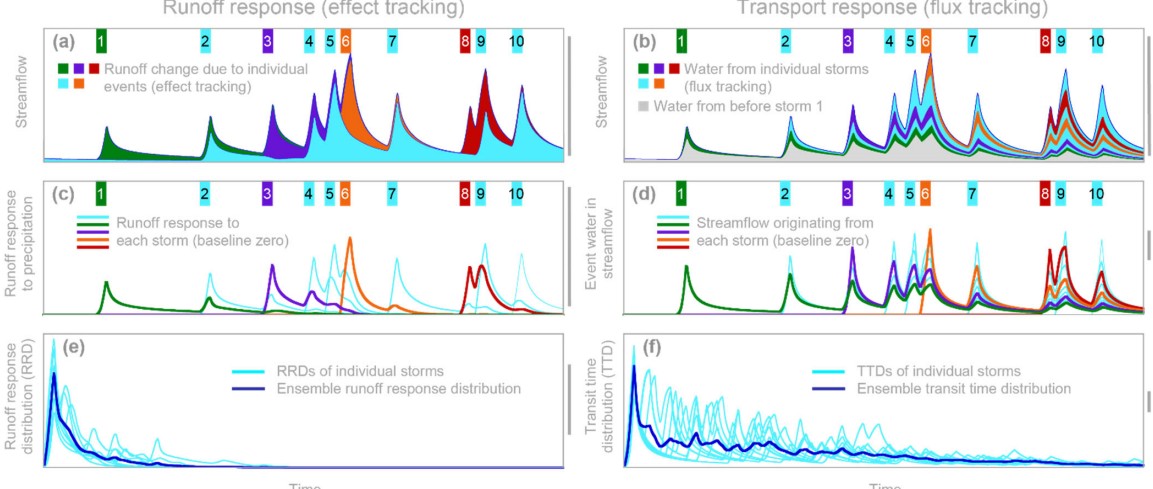

**Figure 1. Effects of 10 rainstorms on future runoff (left panels) compared to future streamflow composition (right panels), illustrated using the nonlinear two-box benchmark model of Kirchner (2019). In (a), the colored bands show the runoff response to the corresponding storms, as determined by effect tracking. Effect tracking quantifies the runoff response by how much the discharge time series changes when individual storms are included or excluded from the precipitation time series. Thus (for example) the orange domain shows how much more runoff occurred when storm #6 was included in the precipitation time series, versus when it was excluded (shown by the teal-colored hydrograph). In (b), the colored bands show the streamflow fluxes that originated as precipitation during each storm, as determined by flux tracking (streamflow from precipitation that fell before event #1 is shown in gray). Panels (c) and (d) show the runoff and transport responses, respectively, for all 10 storms, plotted against a baseline of zero so that they can be more clearly compared. Panels (e) and (f) show the same curves, expressed as time since the start of their respective storms. Dark blue curves show the ensemble averages of these runoff response distributions (RRDs, panel e) and transit time distributions (TTDs, panel f). Each horizontal axis shows the same time interval. The vertical scales are identical in (a), (b), and (c), but differ among the remaining panels to more clearly show the behavior. The gray bars along the right edge of each panel show what the relative sizes of the vertical axes would be if the scales were consistent. Model parameters are $S_{u,\mathrm{ref}} = S_{l,\mathrm{ref}} = 400$ mm, $b_u = b_l = 10$, and $\eta = 0.5$. Each of the 10 modeled storms are of equal intensity (20 mm h$^{-1}$) and duration (10 h). The model ignores evapotranspiration, so each 200 mm storm causes 200 mm of additional streamflow in the future (a) and adds 200 mm of water that will eventually leave the catchment (b). Thus the total volumes of the colored bands in (a) and (b) are equivalent, but the hydrological response to precipitation (left panels) decays away much faster than the event water is flushed out of the catchment (right panels). Modified, with permission, from Kirchner et al. (2023).**

Two caveats must be kept in mind when interpreting the results shown in Fig. 1. First, theoretically the runoff response and transit time distributions are instantaneous impulse response functions, whereas in Fig. 1 they are shown as responses to discrete storms of significant duration, to permit easy visualization. Second, and more importantly, one must remember that the behaviors shown in Fig. 1 are not data, but instead are hypothetical results derived from a particular simulation model, for purposes of illustration. Are these results nonetheless at least qualitatively relevant to the real world? Decades of tracer studies have shown that streamflow, even during storm events, is often composed of months or years of previous precipitation inputs (e.g., Brown, 1961; Martinec, 1975; Rodhe, 1981; Sklash, 1990; Buttle, 1994; Kirchner et al., 2000; McGuire and McDonnell, 2006; Jasechko et al., 2016), implying that timescales of storage and transport in catchments are





typically much longer than timescales of storm hydrographs. Thus the scenario illustrated in Fig. 1 may actually

underestimate the contrast between timescales of runoff response and timescales of storage and transport.

The contrast between these timescales has been widely overlooked in hydrology textbooks and rainfall-runoff models (Beven, 2012; McDonnell and Beven, 2014). As noted by Kirchner et al. (2023), one reason may be simple pragmatism. Applied hydrology typically focuses on forecasting and managing runoff, and when communities are threatened by flooding,

questions about the age of the flood waters will seem quaintly academic. Instead, the central question in such cases will be how meteorological conditions and catchment processes combine to mobilize so much water so quickly, no matter what its age might be. In the early days of scientific hydrology, such practical considerations, together with the observation that runoff responds quickly to rainfall, often led to an intuitive assumption that storm hydrographs are primarily composed of "new" water from recent rainfall traveling quickly to the stream, often via overland flow (Horton, 1933). This conceptual

model formed the foundation of decades of unit hydrograph studies (beginning with the work of Sherman, 1932), as well as attempts to relate the unit hydrograph to drainage basin geomorphology (e.g., Ross, 1921; Rodriguez-Iturbe et al., 1982; Gupta et al., 1980), as reviewed by Rigon et al. (2015) and Beven (2020).

The assumption that recent rainfall dominates storm hydrographs was subsequently shattered by tracer data showing that

streamflow, even during storm peaks, is often composed mostly of "old" water from precipitation that fell much earlier (e.g., Hubert et al., 1969; Pinder and Jones, 1969; Sklash and Farvolden, 1979; Sklash, 1990; Neal and Rosier, 1990). This discovery has spurred decades of studies using isotopes and conservative chemical tracers to estimate mean transit times in many different catchments (as reviewed by McGuire and McDonnell, 2006). Analyses of multi-year tracer time series revealed that their spectra exhibit fractal scaling, implying that catchment transport is highly dispersive and transit time

distributions will often have sharp short-time peaks and long tails, with important implications for contaminant transport (Kirchner et al., 2000, 2001; Godsey et al., 2010; Kirchner and Neal, 2013).

In the past decade, catchment tracer behavior has also been re-interpreted in terms of storage selection (SAS) functions, which model the relationship between the age distribution of water leaving the catchment (the transit time distribution) and

the age distribution of water in storage (the residence time distribution) (Botter et al., 2010; van der Velde et al., 2012; Benettin et al., 2015b; Benettin et al., 2015a; Harman, 2015; Rinaldo et al., 2015; Berghuijs and Kirchner, 2017; Benettin et al., 2022). These approaches, however, require *a priori* assumptions about the catchment mixing model or the shape of the transit time distribution or SAS function. Those constraints have been loosened by the recent development of ensemble hydrograph separation, a model-independent technique for estimating marginal (i.e., time-averaged) transit time distributions

directly from observational data (Kirchner, 2019; Knapp et al., 2019; Kirchner and Knapp, 2020). By analyzing data subsets reflecting different catchment conditions or meteorological forcings, this technique can be used to quantify, directly from





data, how transit time distributions respond to variations in factors such as rainfall intensity, flow regime, and antecedent wetness.

Amid the intense focus in recent decades on catchment _travel_ times, and on _flux_ tracking in models (e.g., the right-hand side of Fig. 1), questions of catchment _response_ times, and of _effect_ tracking in models (e.g., the left-hand side of Fig. 1) have received much less attention. As a step toward addressing this imbalance, this paper develops and tests a model-independent, data-driven method for quantifying runoff response distributions from widely available precipitation and streamflow time series. This method, Ensemble Rainfall-Runoff Analysis (ERRA), is made possible by recent mathematical

developments in the estimation of impulse response functions for nonlinear, nonstationary, and heterogeneous systems (Kirchner, 2022, hereafter denoted as K2022). ERRA characterizes hydrologic behavior using runoff response distributions (as shown in the left-hand side of Fig. 1), as a counterpart to Ensemble Hydrograph Separation (Kirchner, 2019; Kirchner and Knapp, 2020), which characterizes transport behavior using transit time distributions (as shown in the right-hand side of Fig. 1).

**1.2    ERRA versus unit hydrographs**

   At its core, ensemble rainfall-runoff analysis is based on least-squares deconvolution of stream discharge time series by precipitation time series. ERRA's heritage thus reaches back to unit hydrograph methods, which have a long history in hydrology (e.g., Sherman, 1932; Snyder, 1955; Dooge, 1973; Bruen and Dooge, 1992), but it differs from conventional unit hydrograph approaches in several important ways. First, classical unit hydrograph methods invoke what Hewlett and

Hibbert (1967) termed "completely arbitrary" assumptions in order to separate the hydrograph into storm runoff and baseflow, and they invoke further assumptions to separate the rainfall time series into effective precipitation and evaporative losses. To avoid making such assumptions, which have substantial influence on the resulting unit hydrographs (Beven, 2012), ERRA analyzes time series of total precipitation (instead of effective precipitation alone) and total discharge (instead of storm runoff alone). Second, the area under conventional unit hydrographs is constrained to equal 1, thus enforcing the

assumption that whatever is defined as storm runoff must equal, on average, whatever is defined as effective precipitation (and making the analysis vulnerable to biases in either of these quantities). ERRA makes no such assumption, and instead allows the area under the runoff response distribution to reflect mass imbalances due to evapotranspiration losses or infiltration to deep groundwater (as well as due to potential biases in discharge and precipitation measurements). Third, in contrast to many unit hydrograph methods, ERRA does not require defining and isolating individual events, but instead

assimilates information from the entire time series of precipitation and streamflow. That is, it analyzes how streamflows at every point in time (not just during pre-defined "events") are correlated with precipitation – or lack thereof – over previous time steps. In this way it can exploit the information contained in differences in streamflows that follow different precipitation patterns, including those that follow no precipitation at all. Indeed, a key objective of ERRA is to quantitatively understand how differences in precipitation patterns, and ambient conditions, shape the relationship between



precipitation and subsequent streamflows. Fourth, whereas unit hydrograph methods primarily seek to predict storm
      hydrographs, ERRA does not focus on hydrograph prediction per se, but rather on characterizing the magnitude and timing
      of rainfall-runoff relationships, and on quantifying how they depend on precipitation intensity, ambient conditions, and
      catchment characteristics. That is, its goal is primarily data-based characterization of hydrologic response, rather than
      prediction of the hydrograph per se.


      Fifth, and perhaps most importantly, whereas conventional unit hydrograph methods assume that the effects of rainfall on
      runoff are linear, stationary, and homogeneous, ERRA is specifically designed to characterize and quantify the nonlinearity,
      nonstationarity, and heterogeneity of runoff response. For example, typical unit hydrograph methods are driven by a single
      whole-catchment precipitation time series and thus implicitly assume either that the same precipitation falls everywhere or
that its effects on runoff are spatially homogeneous (but there are exceptions, e.g. Kothyari and Singh, 1999). By contrast,
      through a combination of deconvolution and demixing methods, ERRA can distinguish and quantify how runoff responds to
      precipitation measured at different points on the landscape, even when those individual hydrologic responses are overprinted
      on one another at the catchment outlet (see Sect. 2.3 below, and Sect. 3 of K2022). Unit hydrograph methods are also based
      on the premise of linear superposition, such that the response to $x$ units of rain falling at time $t$ is assumed to be $x$ times the
response to a single unit of rainfall at time $t$. By contrast, ERRA recognizes that streamflow may respond nonlinearly to
      variations in rainfall intensity, and uses nonlinear deconvolution methods to quantify that response (see Sect. 3 below, and
      Sect. 5 of K2022). Furthermore, conventional unit hydrograph methods assume that runoff response is stationary, such that a
      given unit of rain always has the same effects on runoff, regardless of when it falls. By contrast, ERRA combines
      deconvolution and demixing methods to explicitly quantify how runoff responses vary with ambient conditions, even if those
runoff responses are overprinted on one another (see Sect. 5 below, and Sect. 4 of K2022).

      Here I briefly introduce ERRA and outline some of its potential applications. The mathematical foundations underlying
      ERRA have previously been documented and benchmark-tested in K2022, and those results will only be briefly summarized
      here. Instead, the purpose of the present contribution is to outline several applications through proof-of-concept
demonstrations, thus illustrating the potential of the technique. Software is provided to perform the necessary calculations,
      including uncertainty analysis, in the open-source programming environment R.

## 2   Characterizing spatially heterogeneous hydrological response via deconvolution and demixing

### 2.1   Runoff response distributions (RRDs) as measures of hydrological response

      In their simplest forms, runoff response distributions like the dark blue curve in Fig. 1e can be interpreted as a convolution
kernel that, when convolved with precipitation, yields streamflow. In continuous time this convolution can be expressed as



$$Q(t) = \int\limits_{\tau=0}^{\infty} \text{RRD}(\tau)\ P(t-\tau)\ d\tau \quad , \tag{1}$$

where $Q$ and $P$ are the rates of streamflow and precipitation, respectively, as continuous functions of time, and the runoff response distribution RRD quantifies their coupling at lag time $\tau$. The process of estimating $\text{RRD}(\tau)$ from time series of $Q$ and $P$ is termed deconvolution. For typical hydrological time series measured in discrete time steps of length $\Delta t$, Eq. (1)

becomes

$$Q_j = \sum_{k=0}^{m} \text{RRD}_k\ P_{j-k}\ \Delta t \tag{2}$$

where $Q_j$ is streamflow at time step $j$, $P_{j-k}$ is precipitation occurring $k$ time steps earlier, $\text{RRD}_k$ is the impulse response of streamflow to precipitation at lag $k$, and $m$ is the maximum lag being considered. It is important to keep in mind that in ERRA, Eqs. (1)-(2) are assumed to hold over all time $t$ or over all time steps $j$ (or, in practice, over long spans of time $t$ or

large ensembles of time steps $j$). As explained in Sect. 1.2 above, no attempt is made to isolate individual events for analysis. To estimate the runoff response distribution, it would appear to be straightforward to re-cast Eq. (2) as the multiple linear regression equation

$$Q_j = \sum_{k=0}^{m} \beta_k\ P_{j-k}\ +\ \alpha\ +\ \varepsilon_j \tag{3}$$

(see also Eq. 4 of K2022), where $\text{RRD}_k$ is estimated by $\beta_k/\Delta t$, the constant term $\alpha$ accounts for persistent biases or mass

imbalances, and the residuals $\varepsilon_j$ capture any time-varying errors. However, conventional least-squares regression techniques assume that the residuals $\varepsilon_j$ are temporally uncorrelated white noise, whereas in practice, streamflow estimation errors typically have both autoregressive and moving-average characteristics. These autoregressive moving average (ARMA) errors can arise from several sources. Measurement errors in discharge may be serially correlated. There will also be measurement errors in precipitation, and even if they are not themselves serially correlated, they will be smoothed and

lagged by the same convolution process that smooths and lags the (unknown) true precipitation, leading to correlations in the residuals. Mis-specification of the underlying model, such as its assumption of linearity and stationarity (these assumptions are relaxed later) will also be reflected in serial correlations in the residuals $\varepsilon_j$. Deconvolution in the presence of such ARMA residuals is a non-trivial problem, because the fitted streamflow values will contain serially correlated signals both from the errors $\varepsilon_j$ and from the real-world process that convolves precipitation to generate streamflow, and these signals

need to be distinguished from one another. It is even more challenging to perform such deconvolutions efficiently on realistically large problems, which may involve hundreds of thousands of time steps and hundreds or even thousands of lag times. Readers are referred to Sect. 2 of K2022 for technical details of how this is done, and for benchmark tests demonstrating that ERRA handles this challenge effectively and efficiently (i.e., over three orders of magnitude faster than





the closest built-in R function). ERRA implements this correction for ARMA noise by default, with no intervention required
by users in most circumstances (for an earlier implementation of an analogous approach, see Duband et al., 1993).

One technical detail that is particular to hydrology, and thus not covered in K2022, is that the effective lag time between
precipitation and streamflow, along with the value of $\mathrm{RRD}_k$ at lag $k = 0$, will depend on whether $Q_j$ is the instantaneous
streamflow at the end of time step $j$, or the average streamflow over time step $j$. If $Q_j$ is measured instantaneously at the end
of each time step, the average lag between rainfall and its effect on streamflow is $(k + 0.5)\Delta t$, and $\mathrm{RRD}_k = \beta_k/\Delta t$ for all $k$.
But if $Q_j$ is averaged over each time step, during lag step zero ($k = 0$), streamflow will only reflect the effects of rain that
falls beforehand, and not later during the same time step. Thus streamflow at the end of the time step will reflect all of the
rain that fell during it, while streamflow at the beginning of the time step will reflect none, and streamflow at the middle of
the time step will reflect half. Integrating over the time step yields the result that when lag $k = 0$, $\beta_{k=0}$ is reduced by half so
RRD must be estimated as $\mathrm{RRD}_{k=0} = 2\,\beta_{k=0}/\Delta t$, and the average lag time linking the effects of precipitation to streamflow
is $\Delta t/3$. For lags $k > 0$, $\mathrm{RRD}_k = \beta_k/\Delta t$ and the effective lag time is $k\Delta t$.

### 2.2    Whole-catchment runoff response at Roanoke River

Here I illustrate the estimation of runoff response distributions using the Roanoke River catchment, a 995 km$^2$ drainage basin
with mixed land use lying between Blacksburg and Roanoke, Virginia, USA (Fig. 2). I extracted hourly precipitation time
series for 2006-2022 from the aviation weather records at two airports located ~40 km apart, just beyond the eastern and
western catchment boundaries (Roanoke Regional Airport in Roanoke, and Virginia Tech/Montgomery Airport in
Blacksburg, respectively), and aggregated hourly streamflows for the same period from 15-minute USGS data from USGS
gauge 02055000 in downtown Roanoke (Fig. 2). Figure 3 shows one year of these hourly measurements. The Roanoke and
Blacksburg precipitation time series are imperfectly synchronized, with a Pearson correlation of 0.3 on an hourly timescale
and 0.7 on a daily timescale. Some precipitation events occur nearly simultaneously at both stations, but the Blacksburg
time series contains some precipitation events that were not observed at Roanoke and vice versa, and events occurring at
both locations often differ in magnitude or timing between them. From Fig. 3 one can also see that streamflow responds
most strongly to precipitation that is recorded simultaneously, or nearly so, at both weather stations. This makes sense
because such events are likely to have entailed widespread precipitation over most of the catchment.

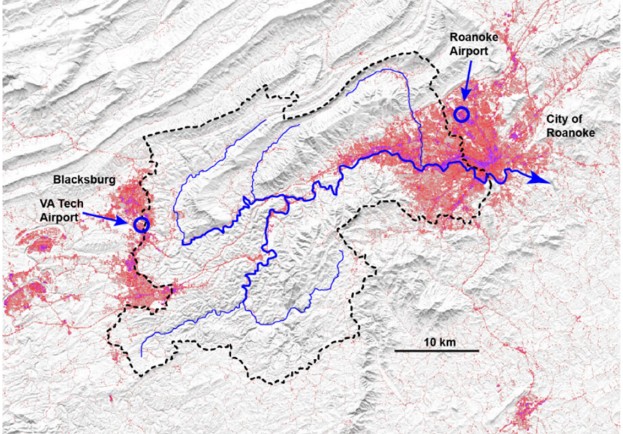


**Figure 2. Map of the 995 km² Roanoke River catchment (Virginia, USA; perimeter shown by black dashed line), with major stream channels (blue lines) and location of airport rain gauges (blue circles). Hillshade is from the USGS National Map Viewer (https://apps.nationalmap.gov/viewer/), with red and purple colors indicating impermeable surfaces in the National Land Cover Database (Dewitz, 2021).**


**Figure 3. One year of hourly precipitation at Roanoke and Blacksburg airports and hourly streamflow at Roanoke gauge. Note that the streamflow axis is expanded 25x relative to the precipitation axes.**

In a typical catchment-scale analysis with multiple weather stations, the precipitation time series are usually averaged

together to yield a single catchment-averaged input. Figure 4a shows the runoff response distribution (RRD) generated by using ERRA (Eq. 3, with ARMA noise correction as described in Sect. 2 of K2022) to deconvolve hourly Roanoke River streamflow by the average of Roanoke and Blacksburg hourly precipitation.





As the name implies, the RRD quantifies how the catchment's streamflow response is distributed over time, per unit of

precipitation. Because precipitation and streamflow are measured in the same units (mm h$^{-1}$), the RRD has dimensions of

time$^{-1}$. If the catchment's response were linear and stationary (time-invariant), the RRD would be a complete description of

its behavior; that is, convolving the precipitation time series with the RRD would yield the streamflow time series. In the

more realistic case of a catchment that is approximately linear but nonstationary, the RRD approximates the ensemble

average streamflow response to precipitation. (If the catchment's runoff response scales nonlinearly with precipitation

intensity, its average is best approximated by a precipitation-weighted average RRD, as described in Sect. 3.4 below.)

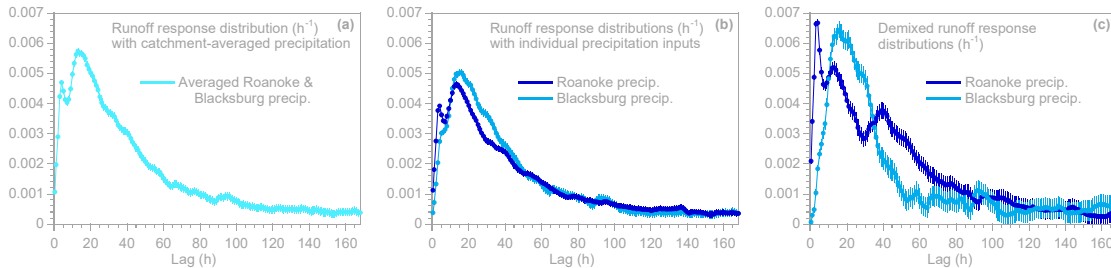

**Figure 4. ERRA runoff response distributions calculated by deconvolving Roanoke River streamflow by different precipitation inputs. In (a), streamflow is deconvolved by the average of precipitation measured at Roanoke and Blacksburg. In (b) streamflow**
**is deconvolved by either Roanoke precipitation alone (dark blue) or Blacksburg precipitation alone (light blue). In (c), by contrast, deconvolution and demixing are used to jointly determine the effects of Roanoke and Blacksburg precipitation on streamflow. This deconvolution/demixing analysis reveals that the effects of Roanoke and Blacksburg precipitation are markedly different. Error bars show one standard error.**

The RRD shown in Fig. 4a shows that the peak streamflow response occurs approximately 15.4±0.9 hours after precipitation

falls, but the streamflow response curve is broad, with a width-at-half-maximum of 39.2±1.6 hours. The integral under the

curve yields an effective runoff coefficient of 0.286±0.001 for all streamflow responses shorter than the maximum lag (here

168 hours = 1 week). When compared to the long-term ratio of 0.345±0.005 between average streamflow (0.042 mm h$^{-1}$)

and average precipitation (0.122 mm h$^{-1}$), this runoff coefficient implies that over 80% of streamflow leaves the catchment

within 1 week of the precipitation that triggered it, with longer-term baseflow comprising the remainder. (Note that this

should not be interpreted as "over 80% of streamflow leaving the catchment within 1 week following the precipitation it

originated from." Nor should the runoff coefficient of 0.289 be interpreted as "29% of precipitation leaving the catchment as

streamflow within 1 week." Hydrometric data reflect the coupling between precipitation and streamflow but do not trace the

movement of water itself. As noted in Sect. 1.1 above, decades of tracer data from many different catchments show that

stormflow is often dominated by precipitation that fell weeks, months, or even years before, and has been stored within the

catchment ever since.)





A notable feature of Fig. 4a is the sharp spike in the runoff response distribution at a lag of roughly 4 hours. Such a rapid and brief runoff response could potentially be generated by rain falling on the Roanoke metropolitan area and being routed rapidly over impermeable surfaces (denoted in red and purple in Fig. 2). One might attempt to gain some insight into this
possibility by deconvolving streamflow by Roanoke precipitation alone, and comparing the resulting runoff response distribution (dark blue curve in Fig. 4b) with one derived by deconvolving streamflow by Blacksburg precipitation alone (light blue curve in Fig. 4b). This comparison appears to imply that Roanoke precipitation triggers a small additional short-term runoff response, but that otherwise the runoff responses to Roanoke and Blacksburg precipitation are similar.

Such an interpretation would be naïve, however, because deconvolving streamflow by Roanoke precipitation does not measure the effects of precipitation falling on Roanoke alone. Instead, it measures the effects of any precipitation, falling anywhere in the catchment, that is correlated with the precipitation measured at Roanoke. In principle, of course, the ideal solution would be to obtain precipitation and streamflow measurements in subcatchments that isolate particular landscapes of interest, such as the Roanoke metropolitan area. But such subcatchment gauging data are unavailable in many situations
like this one. How can we separate the effects of Roanoke and Blacksburg precipitation on streamflow, if we only have streamflow measurements that integrate over the entire catchment?

### 2.3 Deconvolution and demixing of multiple precipitation inputs

The Roanoke and Blacksburg precipitation time series are correlated with one another, and their effects are not only convolved forward through time but are also mixed together at the catchment outlet. This is a common problem in rainfall-
runoff analysis at the catchment scale: precipitation falling in different parts of the landscape with different hydrological properties may generate different streamflow responses, which then may be lagged and dispersed differently on their way to the basin outlet. Estimating the streamflow response to individual inputs therefore combines a deconvolution problem (one must un-scramble each input's temporally overlapping effects) and a demixing problem (one must separate the different inputs' effects from one another). As outlined in Sect. 3 of K2022, this deconvolution-plus-demixing problem can be
approached by analysing the effects of the two precipitation inputs jointly, as follows:

$$Q_j = f_A \sum_{k=0}^{m} \mathrm{RRD}_{A,k}\, P_{A,j-k}\, \Delta t \quad + f_B \sum_{k=0}^{m} \mathrm{RRD}_{B,k}\, P_{B,j-k}\, \Delta t \; , \tag{4}$$

where $\mathrm{RRD}_A$ and $\mathrm{RRD}_B$ are the runoff response distributions for precipitation inputs $P_A$ and $P_B$, which are assumed to fall on fractions $f_A$ and $f_B$ of the catchment. Equation (4) can be re-cast as the multiple regression equation

$$Q_j = \sum_{k=0}^{m} \beta_{A,k}\, f_A P_{A,j-k} + \beta_{B,k}\, f_B P_{B,j-k} + \alpha + \varepsilon_j \; , \tag{5}$$

which is identical to Eq. (3), and is solved identically, except that there are more coefficients to estimate (this approach could of course be generalized to any number of inputs). Equation (5) is analogous to the "multiple-input single-output variable



gain factor model" of Liang et al. (1994), although that approach was applied to flood routing, using flows on tributaries rather than precipitation as inputs. Readers should note that values of the runoff response distributions $RRD_A$ and $RRD_B$ will inevitably be inversely proportional to the mixing fractions $f_A$ and $f_B$; there is no way to determine them independently
without making additional assumptions. This is because a given precipitation input can generate (for example) twice the effect on runoff either by falling on twice as much of the catchment, or by falling on a part of the catchment that is twice as responsive to precipitation. Therefore, although the shapes of the RRDs will be determined by the data, their absolute magnitudes will depend on the assumed values of $f_A$ and $f_B$. Here I assume that $f_A = f_B = 0.5$, which yields the RRDs shown in Fig. 4c.


As Fig. 4c shows, solving jointly for the effects of precipitation at Roanoke and Blacksburg reveals that they are markedly different, in sharp contrast to the results obtained in Fig. 4b by applying each precipitation record separately. Figure 4c reveals a sharp spike in the runoff response to Roanoke precipitation at a lag of just 3.9±0.2 hours (see the dark blue curve in Fig. 4c), presumably reflecting the prevalence of impermeable surfaces in the metropolitan Roanoke area, as well as the
relatively short network flowpaths to the gauging station in downtown Roanoke. By contrast, there is almost no short-term runoff response at Roanoke that is attributable to Blacksburg precipitation (see the light blue curve in Fig. 4c), presumably reflecting the lack of any short flowpaths connecting precipitation falling near Blacksburg with the Roanoke gauging station. The peak runoff response to Blacksburg precipitation is delayed (17.5±0.8 hours) and relatively broad, presumably also reflecting the relative scarcity of impermeable surfaces within the catchment near Blacksburg (Fig. 2). The runoff response
to Roanoke precipitation also shows a broad decline over lags ranging from about 15 to 60 hours, presumably reflecting longer, slower flowpaths to the gauging station, including via tributaries that flow westward before joining the main stream and flowing back eastward to Roanoke (see Fig. 2). A 50-50 mixture of the two response distributions shown in Fig. 4c almost exactly reproduces the response distribution to catchment-averaged precipitation shown in Fig. 4a. Thus Fig. 4c implies that the short-term peak in the runoff response shown in Fig. 4a is generated by precipitation falling near Roanoke,
and the broader, later peak is generated by precipitation falling near both Roanoke and Blacksburg.

These results have the practical implication that storms delivering the same catchment-averaged precipitation can yield very different hydrographs, depending on how much of that precipitation falls near Roanoke versus Blacksburg. Runoff responses to storms can also differ markedly, depending on whether they move from southwest to northeast (such that the
runoff peaks from Blacksburg and Roanoke precipitation tend to coincide), or from northeast to southwest (increasing the separation between the arrival times of the runoff peaks at the outlet). More generally, the analysis presented here demonstrates that whenever one has precipitation records for different parts of a catchment, one can quantify their individual effects on runoff even if the individual subcatchments are not separately gauged. In this way, one can explore how network routing and local variations in catchment characteristics affect runoff dynamics, directly from data, without positing a
physical model and without subcatchment gauging data.



### 3 Quantifying nonlinearities and thresholds in runoff response via nonlinear deconvolution

### 3.1 Introduction: nonlinear deconvolution

It has long been recognized that storm runoff often responds more-than-proportionally to changes in precipitation intensity.
For example, Fig. 5 shows unit hydrographs estimated for 5 brief (10-15 minute) bursts of rainfall in an agricultural catchment in Illinois (data of Minshall, 1960). From Fig. 5 one can see that unit hydrograph peaks are both higher and earlier for higher-intensity storms in this small (11 ha) catchment. Conventional deconvolutions, and unit hydrographs in particular, are inconsistent with the behavior shown in Fig. 5 because they assume that outputs scale proportionally to inputs, leading Minshall (1960) to conclude that a single unit hydrograph cannot adequately characterize the runoff response to
different precipitation intensities, at least in small catchments like those that he studied.

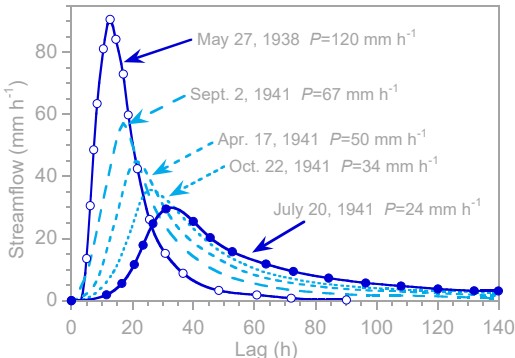

**Figure 5. Unit hydrographs estimated by Minshall (1960) for 10-15 minute periods of different rainfall intensities in an 11 ha catchment.**

Here I outline an approach to characterizing this type of nonlinear runoff response, based on the nonlinear deconvolution method presented in Sect. 5 of K2022. The core of the problem is that one cannot directly observe a catchment's response to (for example) a single hour of precipitation at a given intensity, because these responses are convolved forward through time, overlapping with the responses to other hours of precipitation at other intensities. To handle this problem, I adapt the methods outlined in Sect. 2 above by replacing each coefficient of the runoff response distribution with a function of the
precipitation rate. If the RRD at each lag is a function of precipitation intensity, Eq. (2) becomes

$$Q_j = \sum_{k=0}^{m} P_{j-k} \, \text{RRD}_k(P_{j-k}) \; \Delta t \quad , \tag{6}$$





(see also Ding, 2011), where the parentheses indicate functional dependence rather than multiplication. Because rainfall is stochastic and its distribution is highly skewed, the precipitation intensity $P_{j-k}$ will be inherently dependent on the duration $\Delta t$ over which it is measured. The runoff response distribution $\mathrm{RRD}_k$ may likewise vary with the time step $\Delta t$. It may also

be highly uncertain when precipitation intensity and the resulting runoff response are close to zero. For these reasons, and to more clearly visualize the functional relationship between precipitation intensity and streamflow response, it is useful to define a nonlinear response function $\mathrm{NRF}_k$ as the product between the precipitation-intensity-dependent $\mathrm{RRD}_k$ and the precipitation rate $P$:

$$\mathrm{NRF}_k\big(P_{j-k}\big) = P_{j-k}\,\mathrm{RRD}_k\big(P_{j-k}\big)\ , \tag{7}$$

where the parentheses indicate functional dependence rather than multiplication. Combining Eq. (7) and Eq. (6) yields

$$Q_j = \sum_{k=0}^{m} \mathrm{NRF}_k(P_{j-k})\,\Delta t \tag{8}$$

where $Q_j$ is streamflow at time step $j$, $P_{j-k}$ is precipitation occurring $k$ time steps earlier, $\mathrm{NRF}_k$ is the nonlinear response of streamflow to precipitation that falls at a rate $P_{j-k}$ and lasts for a time step of $\Delta t$, $m$ is the maximum lag being considered, and the parentheses indicate functional dependence rather than multiplication. From Eqs. (6) and (7), one can see that the

units of the NRF will be mm h$^{-2}$, consistent with its interpretation as the rate of streamflow expected to result at a given time lag $k$ from precipitation falling at a rate $P$, per unit time that this precipitation falls. In cases where the time step is equal to the time unit (for example, where precipitation and streamflow are measured in mm h$^{-1}$, over time steps of 1 h), the time step of $\Delta t$ will equal 1 and the NRF will be numerically (though not dimensionally) equal to the increment of streamflow resulting from one time step of precipitation at rate $P$.


The question now becomes how to estimate the nonlinear response function $\mathrm{NRF}_k$. We want to avoid needing to specify the form of this function, and instead allow it to be determined from the data. The approach adopted here and in K2022, as illustrated schematically in Fig. 6a, is to approximate $\mathrm{NRF}_k$ using a continuous piecewise-linear "broken stick" model, with linear segments intersecting at knots that correspond to user-defined precipitation intensities. Dividing the precipitation axis

of Fig. 6 into $n_\kappa$ segments between knots $\kappa_\ell$ allows precipitation intensity to be re-expressed as a vector of values $P'_\ell$ that quantify how much of each segment lies at or below any given value of $P$:

$$P = \sum_{\ell=1}^{n_\kappa} P'_\ell, \qquad P'_\ell = \begin{cases} 0 & \text{if} & P < \kappa_{\ell-1} \\ P - \kappa_{\ell-1} & \text{if} & \kappa_{\ell-1} \le P < \kappa_\ell \\ \kappa_\ell - \kappa_{\ell-1} & \text{if} & P \ge \kappa_\ell \end{cases} \tag{9}$$
$$= \max\big(0, \min(P - \kappa_{\ell-1},\ \kappa_\ell - \kappa_{\ell-1})\big)$$

The nonlinear response function $\mathrm{NRF}_k$ thus becomes the sum of these individual precipitation components, each multiplied by the slopes $\beta'_{\ell,k}$ of the corresponding broken-stick segments (and rescaled by the time interval $\Delta t$),





$$\text{NRF}_k(P) \approx \sum_{\ell=1}^{n_\kappa} \beta'_{\ell,k} \, P'_\ell / \Delta t \quad , \tag{10}$$

and the slopes $\beta'_{\ell,k}$ can be estimated by a linear regression equation that is formed by combining Eqs. (10) and (8),

$$Q_j \approx \sum_{k=0}^{m} \sum_{\ell=1}^{n_\kappa} \beta'_{\ell,k} \, P'_{\ell,j-k} + \alpha + \varepsilon_j, \quad . \tag{11}$$

For the methodological details underlying this approach, users are referred to Sect. 5 of K2022, and for practical

implementation details, including different options for setting the knot values $\kappa_\ell$, they are referred to the documentation for

the ERRA script itself.

It bears emphasis that this approach differs from conventional transfer function models that use a nonlinear transformation to

convert total rainfall to effective rainfall, and then estimate a transfer function to route this rainfall through the catchment

(e.g., IHACRES; Jakeman et al., 1990). The present approach, by contrast, estimates nonlinear relationships like those

shown in Fig. 6 separately for each lag between 0 and $m$; in the conceptual framework of transfer function models, this

means that the effective rainfall can vary with lag time.

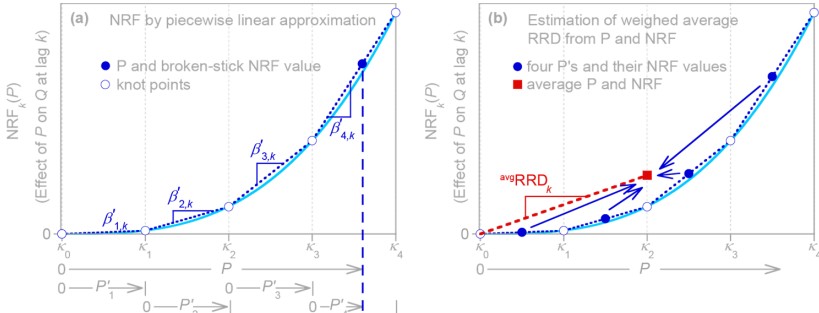

**Figure 6. Nonlinear response function (NRF) estimation by piecewise-linear broken-stick approximation. (a) Estimation of the**
**nonlinear dependence of $Q$ on precipitation intensity $P$ at a specified lag $k$. Splitting the precipitation value $P$ (solid circle) into**
**sub-ranges $P'_\ell$ between user-specified knots $\kappa_\ell$ (open circles) facilitates the fitting of slopes $\beta'_{\ell,k}$ between these knots by linear**
**regression, yielding a nonparametric piecewise-linear continuous curve (see Eqs. 9-10). (b) Estimation of the weighted-average**
**runoff response distribution ($\text{RRD}_k$) as the ratio between the means of the NRF and $P$ (red square) over all time steps (4 example**
**values of $P$ and NRF shown by solid blue circles).**


### 3.2    Profiles of nonlinear response at Saco River

Figure 7 illustrates the approach outlined above, using the Saco River as a test case. At the Conway, NH stream gauge, the

Saco River drains 997 km² of the White Mountains. I aggregated USGS 15-minute discharge measurements from the





Conway gauge to hourly intervals, to correspond with hourly estimates of catchment-averaged precipitation that are available

from MOPEX (Duan et al., 2006). To minimize the effects of snow, the months of November through April were excluded. In all, about 16 years of overlapping hourly precipitation and discharge data are available, spanning 1987 through 2003.

Figure 7a shows NRF estimates of Saco River runoff response to 7 ranges of precipitation intensity. Unsurprisingly, the highest precipitation intensities yield the strongest runoff responses and the highest peaks. They also have the largest error

bars, because the precipitation distribution is highly skewed and thus the highest intensity ranges contain relatively few data points. The peak height, calculated from a quadratic fit to all NRF values within 20% of the peak, increases nonlinearly with precipitation intensity, presumably reflecting the effects of interception losses and refilling of near-surface storage (Fig. 7b).

The total runoff volume, calculated by integrating the NRF over the 100-hour range of lag times shown in Fig. 7a, also

increases nonlinearly with precipitation intensity (Fig. 7c). Because the units of the NRF are mm h$^{-2}$, the units of the runoff volume shown in Fig. 7c are mm h$^{-1}$, which is mm of streamflow (here, in the first 100 h following precipitation), per h of precipitation at a given intensity. One could also equivalently consider Fig. 7c to be a plot of one-hour precipitation intensity in mm (on the x axis) versus the total streamflow resulting from one hour of that precipitation within the first 100 hours (on the y axis). The gap between the runoff volume curve and the 1:1 line in Fig. 7c indicates the volume lost to interception and

evapotranspiration, and also to infiltration that does not generate streamflow within 100 hours.

Peak height increases with runoff volume, following a power function with an exponent greater than 1 (Fig. 7d). This more-than-proportional increase in peak height implies that higher precipitation intensities do not amplify runoff response by the same proportion at all lags, but instead amplify runoff response near the peak lag by somewhat larger factors. The runoff

coefficient, calculated as the ratio between the y and x axes of Fig. 7c, increases with precipitation intensity, reflecting the greater relative importance of interception losses and storage deficits at lower precipitation rates (Fig. 7e). The runoff coefficient appears to reach an upper limit of approximately 0.6, suggesting that even at high precipitation rates, interception and infiltration losses may remain significant. Runoff response peaks are narrower and earlier at higher precipitation rates (Fig. 7f). Figures 7b-7f can be considered as profiles of nonlinear hydrological response, and thus as fingerprints of

catchment behavior. Controls on nonlinear hydrological response may potentially be illuminated by comparing these response profiles among streams with different catchment characteristics.



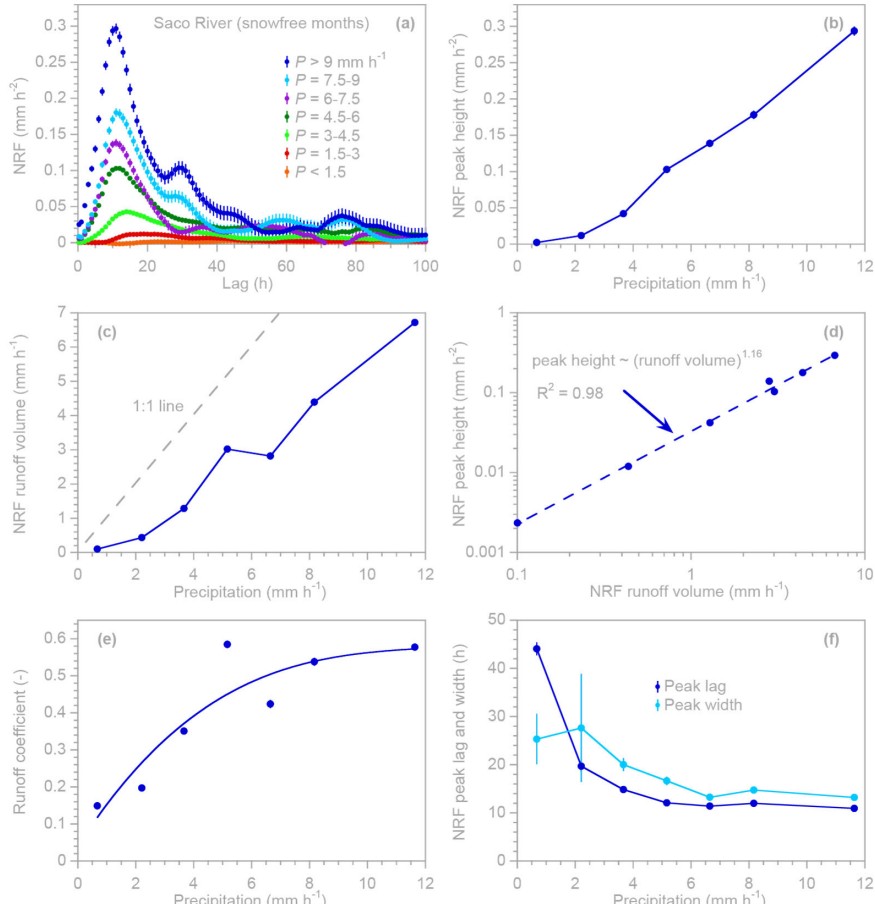

**Fig. 7. Nonlinear rainfall-runoff behavior at Saco River near Conway, NH, USA, inferred from ERRA nonlinear response functions (NRF). (a)** NRF as a function of lag time for specified ranges of precipitation intensity. **(b)** Peak height of NRF as a function of precipitation intensity, showing nonlinearity in peak runoff response. **(c)** Effect of precipitation intensity on NRF runoff volume, measured as the total area under each curve in panel (a) and expressed in mm of streamflow per hour of precipitation at the stated intensity. Runoff volume increases more slowly than precipitation intensity does (i.e., the gap between the blue curve and the gray 1:1 line continues to grow), suggesting that losses to evapotranspiration and/or long-term storage are not constant, but instead increase with increasing precipitation intensity. **(d)** Log-log relationship between NRF peak height and NRF runoff volume, with a scaling exponent of 1.16 indicating that peak height is modestly more sensitive to precipitation intensity than total runoff volume is. Thus as precipitation intensity increases, peak height grows more-than-proportionally relative to average runoff response. **(e)** Runoff coefficient (ratio between NRF runoff volume and precipitation intensity) as a function of precipitation intensity, with an arbitrary smooth curve to guide the eye. **(f)** Lag-to-peak and peak width (at half maximum) as functions of precipitation intensity, showing that NRF peaks are earlier and narrower at higher precipitation rates. Source data span 1987-2003; months from November through April were omitted to exclude effects of snow. Error bars show one standard error, where this is larger than the plotting symbols.



### 3.3    Effects of sampling interval on nonlinear response functions

The stochastic nature of precipitation means that precipitation rates measured over different time intervals will have different distributions. This naturally leads to the question of how the results reported in Fig. 7 might look different, if precipitation rates and runoff responses were measured at different time resolutions.

This question is important because one may want to compare runoff responses at different catchments that have different
sampling frequencies. Furthermore, when the sampling interval is much shorter than the response time of the catchment itself, the runoff time series will be strongly autocorrelated, and so will the residuals of any analysis such as Eqs. (3), (5), or (11). ERRA is designed to handle autocorrelated residuals, but if the autocorrelation is too strong (e.g., lag-1 autocorrelation >0.99), the NRF and RRD may exhibit spurious features and large error bars. These problems can generally be avoided by aggregating the input data over longer and longer time steps until the residual autocorrelation becomes manageable. Thus it
is important to understand how aggregating the underlying time series might affect the results obtained from ERRA.

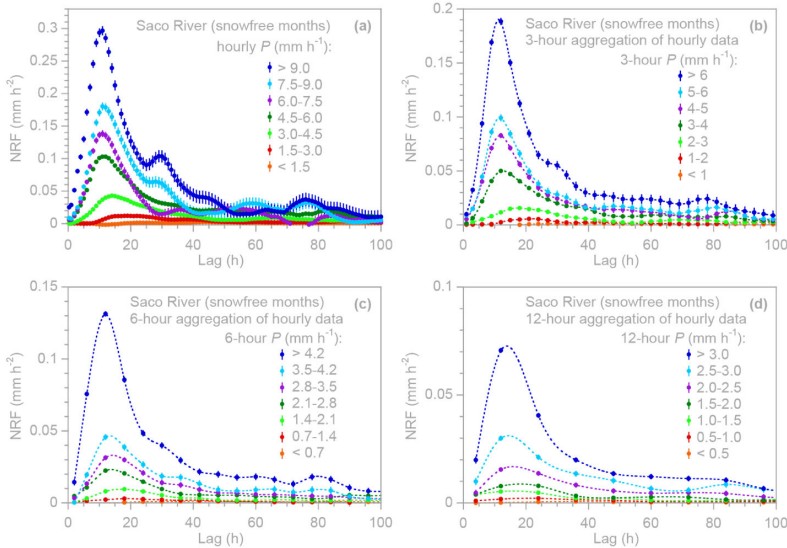

**Fig. 8. Effects of time step aggregation on inferred nonlinear runoff response of Saco River. Panels (a)-(d) show ERRA nonlinear response functions (NRFs) as functions of lag time for specified ranges of precipitation intensity, using hourly precipitation and**
**streamflow time series (a), and the same data aggregated by averaging over 3 hours (b), 6 hours (c), and 12 hours (d). Average precipitation intensity is lower, and runoff responses are correspondingly more muted, when precipitation and discharge are averaged over longer spans of time. Source data span 1987-2003; months from November through April were omitted to exclude any effects of snow. Error bars show one standard error.**

Figure 8 shows the runoff response curves for Saco River, with precipitation and streamflow aggregated over intervals from
1 h to 12 h. The timing of the peak response and the shapes of the curves are similar across the different time scales, but the



precipitation intensities, and thus the NRF values, decrease with increasing time aggregation. However, as Fig. 9 shows, the
underlying relationships between runoff response peak height, runoff volume, peak lag, and precipitation intensity are
generally consistent across the different measurement intervals, with each of the response profiles lying approximately on
top of one another. Longer aggregation time scales, however, inherently lead to smaller ranges of precipitation intensities,

with the result that a smaller part of the response profile is visible. Thus the results obtained from ERRA at different levels
of time step aggregation are consistent with one another, but not equivalent to one another.

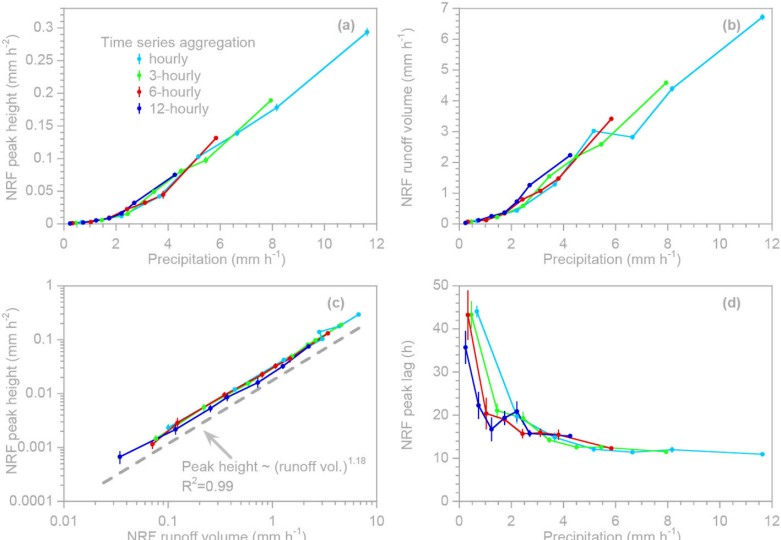

**Fig. 9. Nonlinear responses of peak heights, peak lags, and runoff volumes to variations in precipitation intensity at Saco River,**
**estimated from precipitation and streamflow data aggregated over different time intervals. Time step aggregation reduces**
**average precipitation intensity and damps the resulting nonlinear response functions (NRFs; Fig. 8). However, the nonlinear**
**response curves relating peak heights (a) and runoff volumes (b) to precipitation intensity plot on top of one another at different**
**levels of time step aggregation, as do the power-law relationships between peak heights and runoff volumes (c; dashed gray line is**
**offset from the data for clarity). Time step aggregation has little effect on lag-to-peak (d) at high precipitation intensities, but may**
**have a more noticeable effect at low precipitation intensities. Source data span 1987-2003; months from November through April**
**were omitted to exclude any effects of snow. Error bars show one standard error.**

### 3.4 Average runoff response distribution of nonlinear systems

The NRF quantifies the system's response to one time step of precipitation that falls at an intensity of $P$ (averaged over the
time step). Thus if $P$ and $Q$ are measured in mm h$^{-1}$ at hourly time steps, for example, the units of the NRF will be mm h$^{-2}$.
The NRF is not, however, normalized by precipitation intensity like the RRD is. For any individual value of $P$, one could
straightforwardly estimate the runoff response per unit of precipitation as $\mathrm{NRF}_k(P)/P$. This could be viewed as the runoff





response distribution for a specific range of precipitation intensities, as in Eq. (6), but one should be aware that it may yield highly uncertain results for low-intensity precipitation with small runoff response. A better approach, and the one adopted in

ERRA, is to define the weighted average runoff response distribution as equalling the average of the NRF over all time steps, divided by the average P, as illustrated schematically in Fig. 6b. This will equal the precipitation-weighted average of the runoff response distribution $\mathrm{RRD}_k(P_{j-k})$ over all time steps:

$$^{\mathrm{avg}}\mathrm{RRD}_k = \frac{\sum_{j=1}^{n-k} \mathrm{NRF}_k(P_{j-k})}{\sum_{j=1}^{n-k} P_{j-k}} \Delta t = \frac{\sum_{j=1}^{n-k} P_{j-k}\, \mathrm{RRD}_k(P_{j-k})}{\sum_{j=1}^{n-k} P_{j-k}} \tag{12}$$

This average runoff response distribution will be inherently dependent on the distribution of precipitation intensities,

whenever the system response is nonlinear. Readers should also note that whenever the runoff response is nonlinear, and thus the underlying relationship between NRF and $P$ is curved, the point defining the average $P$ and average NRF (e.g., the red square in Fig. 6b) will not lie along the function $\mathrm{NRF}(P)$, but instead will lie inside of the curve. Thus, in the typical case of an upward-curving NRF, the average NRF will be greater than the value of the NRF evaluated at the average $P$, and thus the weighted average RRD will also be greater than the RRD evaluated at the average $P$. The weighted average RRD is

nonetheless the average runoff response per unit of precipitation (averaged over the nonlinear relationship between precipitation intensity and runoff response), and thus is the closest analogue to the runoff response distribution of a linear system.

One should also be aware that if the system is nonlinear, RRD's calculated via Eqs. (1-3) – that is, without recognition of the

system's nonlinearity – will generally overestimate the precipitation-weighted average RRD (Eq. 12). This overestimation bias arises because precipitation distributions typically have very long upper tails, so the highest-precipitation points (whose leverage scales as roughly $P^2$) have disproportionate influence on the RRD estimate. This bias, which is inherent in all regression-based estimates of unit hydrographs, may be desirable if one wants an estimate that is skewed toward catchment response to high-intensity precipitation. But if one wants to capture the average runoff response of nonlinear systems, the

approach outlined in Eqs. (6-12) will be needed.

Here I illustrate this approach using rainfall-runoff data from five rivers in the southeastern US that exhibit different degrees of nonlinearity and markedly different response times (Fig. 10). As in Sect. 3.2 above, I aggregated 15-minute discharge measurements from USGS gauges on each of these rivers to hourly intervals, and combined them with hourly estimates of

catchment-averaged precipitation that are available from MOPEX (Duan et al., 2006) for the corresponding drainage basins. The resulting time series for the five sites span between 13 and 18 years of hourly data.

As Fig. 10 shows, precipitation-weighted average RRDs calculated via Eq. (12) for all five sites are less strongly peaked (dark blue symbols, left panels) than unweighted RRDs (light blue symbols, left panels) calculated via Eqs. (1)-(3). It may





seem counterintuitive that the unweighted RRDs are more strongly peaked, and that accounting for the nonlinearities in runoff response and weighting by precipitation dampens the RRD peaks rather than sharpening them. But it is important to remember that as noted above, because the mean $P$ is close to zero, each point's leverage in Eq. (3) is approximately $P^2$, so the "unweighted" RRDs are implicitly weighted by the square of precipitation. The resulting overestimation bias is greatest at sites like Clinch River or the South Fork New River, where the NRF peak is a strongly nonlinear function of precipitation

intensity. By quantifying this nonlinearity and explicitly weighting by $P$, the approach of Eqs. (9)-(12) corrects the overestimation bias that arises in simpler approaches like Eqs. (1)-(3), and in similar regression-based approaches to unit hydrograph estimation.

Figure 10 also demonstrates a wide range of hydrologic response timescales among the five catchments, with the peak runoff response in the weighted average RRD ranging from just over three hours (for the Northeast Branch of the Anacostia River at Riverdale, Maryland, draining 189 km$^2$ of a mostly suburban landscape north of Washington DC) to 48 hours (for the Clinch River above Tazewell, Tennessee, draining 3818 km$^2$ of mostly forests and farmlands of the Appalachian Mountains). Notably, only the slowest runoff responses among these sites would be captured by daily time series, which are widely used for hydrological analysis and modelling. Thus daily time series, and models that are calibrated to them, may fail to reflect

the rapid dynamics that characterize runoff response in many landscapes.



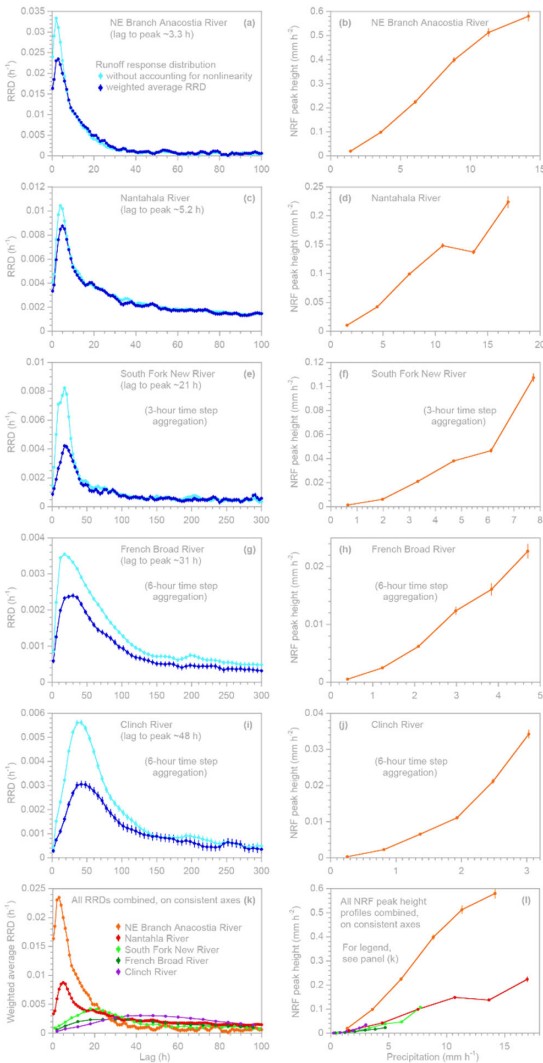

**Figure 10.** Runoff Response Distributions (RRDs) calculated with (dark blue) and without (light blue) accounting for nonlinear response to precipitation intensity (left panels), and profiles of Nonlinear Response Function (NRF) peak height as a function of precipitation intensity (right panels), for five mesoscale river basins in the southeastern US. Basins are the Northeast Branch of the Anacostia River at Riverdale, MD (a, b: 189 km$^2$), Nantahala River near Rainbow Springs, NC (c, d: 134 km$^2$), South Fork New River near Jefferson, NC (e, f: 531 km$^2$), French Broad River at Asheville, NC (g, h: 2448 km$^2$), and Clinch River above Tazewell, TN (i, j: 3818 km$^2$). Weighted average RRDs, calculated via Eq. (12) to take account of nonlinear response (dark blue, left panels), are less strongly peaked and peak a bit later than RRDs calculated without taking nonlinearity into account (light blue, left panels). Note that the axis scales differ substantially among the different panels. All sites' weighted average RRDs and NRF peak height profiles are shown on consistent axes in panels (k) and (l). At several sites with delayed and long-lasting runoff response, time steps were aggregated to 3 or 6 hours to prevent the residual autocorrelation from becoming excessive. Error bars indicate one standard error.





### 3.5 Nonlinear storage-discharge relationships

It should be clear that here in Sect. 3, the term "nonlinear" has been used to refer to catchments that respond nonlinearly to variations in precipitation intensity, for example as shown in Figs. 7-9. Elsewhere in the hydrological literature, the term "nonlinear" has also been used to refer to catchments in which streamflow is a nonlinear function of catchment storage:

$$Q = f(S) \ , \ \frac{\mathrm{d}S}{\mathrm{d}t} = P - E - Q \qquad (13)$$

where $Q$, $S$, $P$ and $E$ (discharge, storage, precipitation, and evapotranspiration, respectively) are all functions of time, and the function $f$ may have any general nonlinear form. Data-driven streamflow prediction models for such systems can be fitted using Volterra series (e.g., Amorocho, 1967; Kothyari and Singh, 1999), which represent streamflow as a linear function of a multidimensional array of past inputs. But even a second-order Volterra series requires $m \cdot \left(1 + \frac{m+1}{2}\right)$ coefficients and a third-order series requires $m \cdot \left(1 + \frac{m+1}{2} \cdot \left(1 + \frac{m+2}{3}\right)\right)$ coefficients, so the coefficients can rapidly become too numerous to

estimate accurately. This overfitting problem can be handled using regularization methods (Kothyari and Singh, 1999), or by approximating the (multidimensional) response function with polynomials such as Meixner functions or Chebyshev polynomials (Amorocho and Brandstetter, 1971), which can greatly reduce the number of coefficients to be estimated, at the cost of obscuring sharp features in the response function. Whatever the specific calculation procedure, however, a Volterra series is difficult to interpret in terms of impulse response (i.e., a runoff response distribution). From the impulse response

perspective that is the focus of this paper, a nonlinear system such as Eq. (13) is more properly considered to be nonstationary, because even if the coefficients of the function $f$ are constant over time, the response to precipitation will depend not only on input intensity, but also on the value of the storage $S$ (and thus on the prior history of inputs, which is what the Volterra series approach aims to capture). The next section presents methods for quantifying how antecedent wetness and other nonstationary catchment properties affect runoff response to precipitation.

## 4 Quantifying nonstationary controls on runoff response

### 4.1 Introduction

From the impulse response perspective, a catchment's wetness status is a nonstationary property that influences its response to precipitation falling at any moment. Other examples of nonstationary properties influencing runoff response include air temperature (which influences the phase of precipitation and the energy available to drive evapotranspiration), vapor

pressure deficit (which influences how rapidly intercepted precipitation evaporates), and the leaf area index of vegetation (which influences how much precipitation is intercepted in the first place). Thus the same precipitation inputs may generate substantially different runoff responses, depending on the ambient conditions. Precipitation inputs themselves may also





change those ambient conditions (for example, by increasing soil moisture and reducing the vapor pressure deficit), and thus influence catchment response to future precipitation inputs.


The challenge, then, is to characterize how a landscape's response to precipitation inputs will vary, depending on the ambient conditions when those inputs fall. This is both a deconvolution problem (because the lagged effects of precipitation inputs are overprinted on each other) and a demixing problem (because the effects of precipitation falling under one set of ambient conditions are overprinted on the effects of precipitation falling at other times under other ambient conditions). These two

problems can be solved simultaneously via a deconvolution-demixing approach analogous to that presented in Sect. 2.3 above. In a simple case, we might be able to separate the precipitation time steps into two groups according to the ambient conditions when the rain falls (for example, groups A and B corresponding to antecedent wetness above and below a particular value). Then if we use $\mathrm{RRD}_{\mathrm{A},k}$ and $\mathrm{RRD}_{\mathrm{B},k}$ to represent the runoff response distributions over lags $k$ resulting from precipitation that falls under wet and dry antecedent conditions, respectively, the discharge time series becomes:

$$Q_j = \sum_{k=0}^{m} \mathrm{RRD}_{\mathrm{A},k}\, P_{\mathrm{A},j-k}\, \Delta t \;\; + \sum_{k=0}^{m} \mathrm{RRD}_{\mathrm{B},k}\, P_{\mathrm{B},j-k}\, \Delta t \;, \tag{14}$$

where $Q_j$ is streamflow at time step $j$, $P_{\mathrm{A},j-k}$ is precipitation falling $k$ time steps earlier under wet conditions (and zero otherwise), and $P_{\mathrm{B},j-k}$ is precipitation falling $k$ time steps earlier under dry conditions (and zero otherwise). Equation (14) can be re-cast as the multiple regression equation

$$Q_j = \sum_{k=0}^{m} \beta_{\mathrm{A},k}\, P_{\mathrm{A},j-k} + \beta_{\mathrm{B},k}\, P_{\mathrm{B},j-k} + \alpha + \varepsilon_j \;, \tag{15}$$

which is identical to Eq. (5) except it lacks the area fractions $f_{\mathrm{A}}$ and $f_{\mathrm{B}}$. Section 4 of K2022 outlines the derivation of this approach and presents several benchmark tests of it. This approach can be combined with the nonlinear deconvolution methods outlined in Sect. 3 above, yielding regression equations of the form

$$Q_j \approx \sum_{k=0}^{m} \sum_{\ell=1}^{n_\kappa} \beta'_{A,\ell,k}\, P'_{A,\ell,j-k} + \beta'_{B,\ell,k}\, P'_{B,\ell,j-k} + \alpha + \varepsilon_j, \tag{16}$$

that quantify the combined effects of variations in antecedent wetness and precipitation intensity. It should be clear that Eqs.
(14)-(16) can be generalized to any number of categories, potentially representing combinations of different ambient conditions (such as, for example, multiple levels of antecedent wetness in both the growing season and the dormant season).

ERRA can automatically separate the precipitation time series according to defined ranges (expressed as either values or percentiles) of nested combinations of multiple variables, and quantify the runoff response across all of these categories
simultaneously, while also correcting for ARMA noise (as outlined in Sect. 2 above and described in more detail in Sect. 2 of K2022).

### 4.2 Proof of concept

Here I illustrate this approach using rainfall-runoff data from the Plynlimon research catchments in Wales (Fig. 11). Hourly

weather data are available from four weather stations, and discharge data are available every 15 minutes from 10 stream gauges with drainage areas ranging from 0.9 to 10.5 km$^2$. For seven of these gauges, records are available for at least 35 years from the mid-1970's through 2010 (Marc and Robinson, 2007); more recent measurements are also available, but here I analyze older data that have been extensively quality-controlled. The climate is generally cool and humid, with annual precipitation averaging roughly 2500-2600 mm per year (Marc and Robinson, 2007); over 70% of days have some

measurable precipitation, and over 24% of days have precipitation totals exceeding 10 mm. The resulting hydrographs are flashy, with many high-flow events each year (Fig. 11b-d).

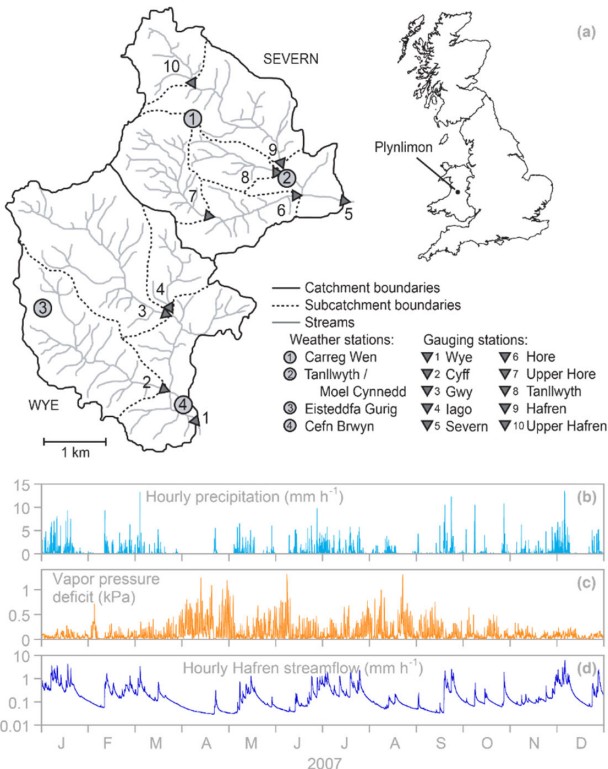

**Figure 11. Map of the Plynlimon research catchments (a), redrawn with permission from Kirchner (2009), and one example year**
**of hourly time series of precipitation (b), vapor pressure deficit (c) and Hafren streamflow (d).**





Using hourly data from 1976 to 2010 for the 3.6 km$^2$ Hafren catchment as a proof-of-concept case (catchment 9 in Figure 11a, with precipitation estimated by averaging measurements at weather stations 1 and 2 in Figure 11a), I tested how runoff responses to precipitation vary between wet and dry antecedent conditions (Figure 12). Because long-term records of soil

moisture or groundwater levels are not available, I use antecedent discharge (ant$Q$, the streamflow measured during the hour before rain falls) as a proxy for antecedent wetness at the catchment scale. Figure 12a shows RRDs estimated via Eq. (15) for five ranges of antecedent discharge, separated by the approximate 40th, 67th, 90th, and 97th percentiles of the Hafren discharge distribution. For the same five ranges of antecedent discharge, Fig. 12b shows weighted average RRDs, estimated via Eq. (12) from nonlinear response functions that account for nonlinear effects of variations in precipitation intensity.


The runoff response shown in Figs. 12a and 12b is markedly larger, and somewhat quicker, under wetter antecedent conditions. Readers may worry that this is an artifact of the use of antecedent discharge as a measure of antecedent wetness. However, the RRDs in Fig. 12 are not determined by discharge itself (which would indeed be correlated with antecedent discharge), but rather by how much discharge changes, depending on how much precipitation falls (see Eqs. 3 and 15). Thus

the effects of antecedent discharge on the RRDs are not artifactual.

One would expect that runoff response under a given antecedent wetness condition may depend on precipitation intensity, and likewise that runoff response for a given precipitation intensity may depend on antecedent wetness. Figures 12c and 12d illustrate the combined effects of antecedent wetness and precipitation intensity on runoff response at Hafren. Figure 12c

shows how the NRF varies with precipitation intensity (shown by the different colors) for two different ranges of antecedent wetness (shown by the main plot, for high ant$Q$, and the inset plot, for moderate ant$Q$). Similarly, Fig. 12d shows how the NRF varies with antecedent wetness (shown by the different colors), for two different ranges of precipitation intensity (9-12 mm h$^{-1}$, shown in the main plot, and 3-6 mm h$^{-1}$, shown in the inset plot). Both Figs. 12c and 12d demonstrate that the shape, scale, and timing of runoff response can depend jointly on both antecedent wetness and precipitation intensity. For

example, one hour of precipitation at an intensity of 9-12 mm h$^{-1}$ can be expected to raise stream discharge by a maximum of ~2 mm h$^{-1}$ at a lag of ~2 hours if it falls under high antecedent wetness conditions (purple curve in the main plot in Fig. 12c or blue curve in the main plot in Fig. 12d), but by only ~1 mm h$^{-1}$ if it falls under moderate antecedent wetness conditions (purple curve in the inset plot in Fig. 12c or green curve in the main plot in Fig. 12d). In other words, Figs. 12c and 12d reveal runoff response that is nonstationary (i.e., dependent on the antecedent wetness in the landscape when rain falls) and

also scales nonlinearly with precipitation intensity.

In such a catchment, any index of runoff response (such as peak height) can be considered as a joint function of antecedent wetness and precipitation intensity. Figure 13 shows the joint influence of antecedent wetness and precipitation intensity on NRF peak height at Hafren. Figure 13a shows that peak runoff response depends nonlinearly on antecedent wetness, and

that it is more sensitive to antecedent wetness at higher precipitation intensities. Providing a perpendicular view of the same



three-dimensional relationship between antecedent wetness, precipitation intensity, and runoff response, Fig. 13b shows that

peak runoff response depends nonlinearly on precipitation intensity, and that it is more sensitive to precipitation intensity at

higher levels of antecedent wetness. These dependencies may be intuitively reasonable to many hydrologists, but what's new

is that they can now be rigorously quantified, directly from data. The shapes of the curves shown in Fig. 13, and their

numerical values, would not be inferable from data by any previous methods of which the author is aware.

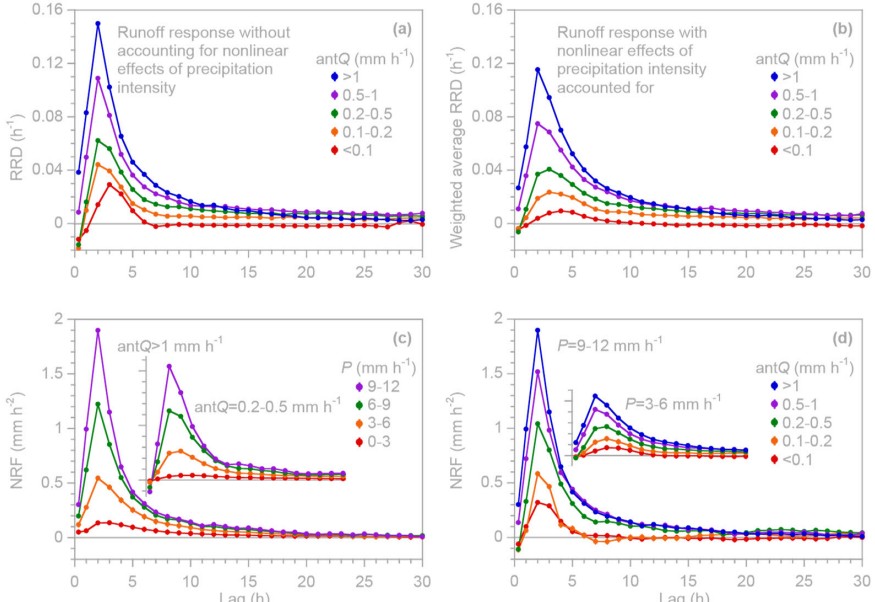

**Figure 12. Runoff responses at Hafren, estimated via Eq. (16) as functions of precipitation intensity $P$ and antecedent wetness (using 1-hour antecedent discharge, ant$Q$, as a proxy). (a) Runoff response distributions (RRDs) for different ranges of ant$Q$, calculated using Eq. (15) without accounting for nonlinear effects of $P$ or co-variation between $P$ and ant$Q$. Peak runoff response**
**is somewhat exaggerated, due to the greater leverage of high precipitation values (see text). (b) Precipitation-weighted average RRDs for different ranges of ant$Q$, with nonlinear effects of $P$, and co-variation between $P$ and ant$Q$, taken into account via Eq. (15). Peak runoff responses in weighted average RRDs (b) are less pronounced relative to unweighted RRDs (a), particularly under drier antecedent conditions. (c) Effects of variations in precipitation intensity $P$ (shown by different colors) for two example ranges of ant$Q$ (as proxy for antecedent wetness), illustrating how runoff response under drier antecedent conditions (lower ant$Q$:**
**inset figure) is less pronounced across all ranges of precipitation intensity. (d) Effects of variations in ant$Q$ (shown by different colors, as proxy for antecedent wetness) for two example ranges of precipitation intensity $P$, illustrating how runoff response to lower-intensity precipitation (lower $P$: inset figure) is less pronounced across all ranges of ant$Q$. Insets in (c) and (d) are on the same axis scales as the main figures, but are cropped and offset for compact presentation. Error bars indicate one standard error, where this is larger than the plotting symbols.**


If runoff response depends on both precipitation intensity and antecedent wetness, it is essential to analyze them jointly (as in

Fig. 13) because otherwise, co-variation between them could bias the assessment of both. Antecedent wetness and

precipitation intensity often co-vary, as seasonal weather patterns or large-scale weather systems raise antecedent wetness

and make intense precipitation more likely. At Hafren, for example, between the lowest antecedent wetness category in Fig.





13 (antecedent discharge < 0.1 mm h$^{-1}$) and the highest (antecedent discharge > 1 mm h$^{-1}$), the mean precipitation rate

increases from 0.13 to 1.68 mm h$^{-1}$ and the 90$^{th}$ percentile of precipitation intensity increases from 0.25 to 4.75 mm h$^{-1}$. Thus

if we just compare runoff responses under different levels of antecedent discharge, we will overestimate the influence of

antecedent wetness because higher antecedent wetness tends to be accompanied by more intense precipitation. Conversely,

if we simply compare runoff responses to different rainfall rates, antecedent wetness may be a hidden variable that

exaggerates the effect of precipitation intensity. Accurately assessing the influence of these interdependent drivers requires

analyzing their effects jointly, which is what Eq. (16) is designed to do.

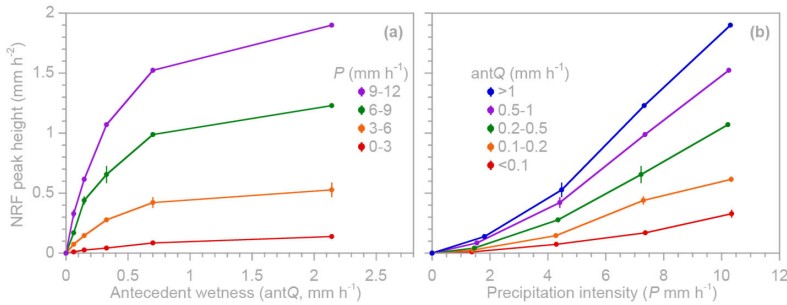

**Figure 13. Peak runoff response at Hafren as a joint function of precipitation intensity *P* and antecedent wetness (using 1-hour**
**antecedent discharge, ant*Q*, as a proxy). (a) Peak height of the nonlinear response function (NRF) as a function of antecedent**
**wetness for four ranges of precipitation intensity (shown by different colors). (b) Peak height as a function of precipitation**
**intensity for five ranges of antecedent wetness (shown by different colors). Error bars indicate one standard error, where this is**
**larger than the plotting symbols.**

The need to jointly quantify the influence of multiple correlated drivers can be illustrated by considering the question of how

vapor pressure deficit (VPD) can influence runoff response to precipitation. One may expect that higher VPD will lead to

faster evaporation of incoming precipitation, and thus to smaller runoff response. But how big is this effect? To exclude the

direct effects of precipitation itself on VPD (whenever rain is falling, the atmosphere is likely to be close to saturation and

thus VPD is likely to be low), here I analyze antecedent VPD (antVPD), the VPD measured during the time step before rain

falls. But even so, one may expect antecedent VPD to be correlated with precipitation rates, partly because hours of rain

tend to follow other hours of rain. Between the lowest 60% of VPD in the Hafren data and the highest 20% of VPD, mean *P*

decreases from 0.44 to just 0.02 mm h$^{-1}$. So clearly, just as with antecedent wetness in Fig. 13 above, accurately assessing

the effects of antecedent VPD and precipitation intensity will require analyzing them jointly, as outlined in Eq. (16). (As we

will see below, it will also require accounting for the co-varying effects of antecedent wetness, but as a cautionary tale I first

show what happens when this confounding factor is overlooked.)





Figure 14 presents a first attempt at analyzing the joint influence of antecedent VPD and precipitation intensity on NRF peak height at Hafren, using five bins of antecedent VPD, each accounting for 20% of the data. (The ranges of precipitation intensity analyzed in Fig. 14 are different from those in Figs. 12 and 13, to ensure sufficient numbers of precipitation events in each combination of precipitation intensity and antecedent VPD.) Both panels of Fig. 14 suggest that the lowest 60% of antecedent VPD has little effect, but that in the upper 40% (and particularly the highest 20%) of antecedent VPD, runoff response is reduced by roughly a factor of 5 relative to the lowest 60% of VPD, even at the highest precipitation intensities (compare the red curve with the blue, purple, and green curves in Fig. 14b).

This effect is surprisingly large, and a moment's reflection gives a good clue why. Periods with high VPD tend to be dry in other ways as well; in particular, weather conditions that lead to higher VPD will also tend to lead to low antecedent wetness in the landscape, and conversely, wetter landscapes promote faster evaporation and thus reduce VPD. Between the lowest 60% of VPD in the Hafren data and the highest 20% of VPD, mean antecedent discharge decreases from 0.27 to 0.11 mm h$^{-1}$; between the same VPD ranges, the 98$^{th}$ percentile of antecedent discharge, corresponding to the wettest landscape conditions, decreases by a factor of 3, from 1.56 to 0.49 mm h$^{-1}$. Thus antecedent wetness may be a confounding variable that amplifies the apparent effect of VPD variations on runoff response.

In such cases, seeing the effect of VPD variations will require analyzing them jointly with variations in both antecedent wetness and precipitation intensity. ERRA can do this seamlessly, setting up Eq. (16) to solve for the nonlinear effects of precipitation intensity in nonstationary systems described by any desired combination of drivers – in this case, two ranges of VPD (the lowest 60% and highest 40%) crossed with five ranges of antecedent discharge). Accounting for both antecedent VPD and antecedent wetness yields markedly different results than those in Fig. 14. As Fig. 15 shows, runoff response in high-VPD conditions (dashed lines, lighter colors) is reduced by roughly 20-30% relative to runoff response in low-VPD conditions (solid lines, darker colors), but broadly follows the same patterns of dependence on antecedent wetness and precipitation intensity. Thus one can infer that, as hypothesized, the large apparent effects of VPD in Fig. 14 are not realistic, but instead arise from the co-variation between VPD and antecedent wetness.

Very humid landscapes like the Welsh setting of the Hafren catchment are hardly ideal for measuring how atmospheric vapor demand influences hydrological behavior. Nonetheless it is reassuring that even here, ERRA can measure such effects, directly from data, by analyzing them jointly with potential confounding factors.

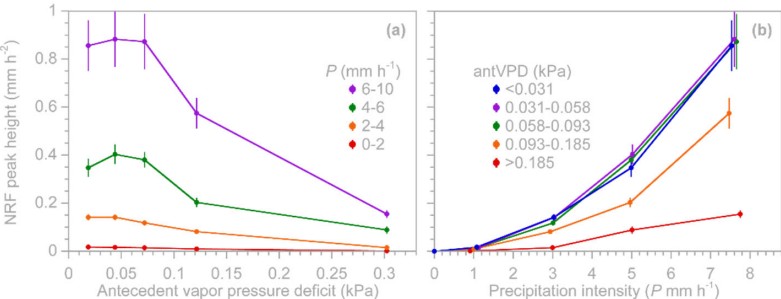

**Figure 14. Peak runoff response at Hafren, as a joint function of precipitation intensity $P$ and 1-hour antecedent vapor pressure deficit (antVPD), without accounting for the co-variation between VPD and antecedent wetness. (a) Peak height of the nonlinear response function (NRF) as a function of vapor pressure deficit for four ranges of precipitation intensity (shown by different colors). (b) Peak height as a function of precipitation intensity for five ranges of antVPD (shown by different colors), corresponding to quintiles of the VPD distribution. Precipitation intensity ranges are more limited than in Figs. 12-13 to provide enough time steps for each combination of $P$ and antVPD. Error bars indicate one standard error, where this is larger than the plotting symbols. High levels of antecedent VPD appear to reduce runoff response by roughly 80%, including when precipitation intensity is high.**

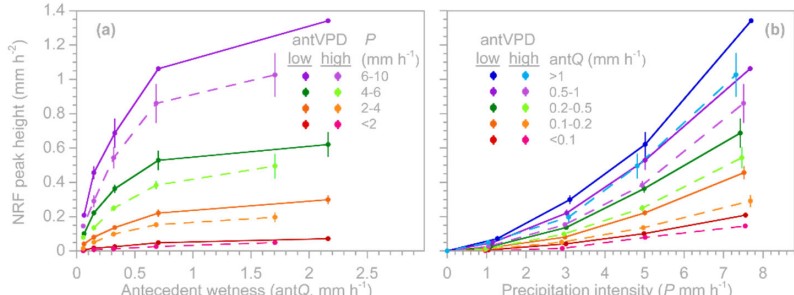

**Figure 15. Peak runoff response at Hafren, as a joint function of precipitation intensity $P$ and antecedent wetness (using 1-hour lagged discharge, antQ, as a proxy), for low antecedent vapor pressure deficit (antVPD) conditions (darker colors and solid lines: lowest 60% of VPD values) and high antVPD conditions (lighter colors and dashed lines: highest 40% of VPD values). (a) Peak height of the nonlinear response function (NRF) as a function of antecedent wetness for four ranges of precipitation intensity (shown by different colors). (b) Peak height as a function of precipitation intensity for five ranges of antecedent wetness (shown by different colors). Precipitation intensity ranges are more limited than in Figs. 12-13 to provide enough time steps in each combination of $P$ and antVPD. Error bars indicate one standard error, where this is larger than the plotting symbols. Jointly accounting for the effects of precipitation intensity and antecedent wetness shows that, all else equal, peak runoff response is roughly 20-30% lower following periods of high VPD (dashed lines) than following low VPD (solid lines). However, peak runoff response following both low and high VPD exhibits similar dependency on antecedent wetness and precipitation intensity.**




## 5    Quantifying multiscale runoff response spanning both short and long lag times

### 5.1    Introduction


Streamflow response to precipitation spans a wide range of time scales, often rising to a peak within minutes or hours, followed by a recession that potentially lasts days, weeks or months. The analysis presented above has focused on short-term responses to streamflow (as most rainfall-runoff analyses do). However, long-tail recession behavior can potentially give interesting insights as well (for example, into catchment storage dynamics; e.g., Kirchner, 2009; Tashie et al., 2019;

Tashie et al., 2020), if it can be accurately quantified. A challenge in recession analysis is that recessions rarely last more than a few days before they are disrupted by new precipitation inputs. In principle, ERRA should be able to filter out the effects of overlapping precipitation inputs, if the runoff response (RRD or NRF) is accurately estimated. But while this approach solves one problem, it creates another, because accurately portraying the long tails of streamflow recession requires accurately estimating the much quicker, and much larger, short-term response to precipitation. How can we capture both the

quick, high peaks of runoff response, and the long tails? If we use time steps that are too long, we will fail to capture the short-term response accurately. But if the time steps are short instead, we will need lots of them, leading to computational inefficiency and also to statistical inaccuracy, because (for example) the effects of an hour of precipitation will be nearly identical at lags of 499 hours and 501 hours, making estimates of coefficients at both lags unreliable.

Since hydrological response changes rapidly soon after rain falls, and slowly thereafter, we would like to approximate that response with a function that does the same thing. But in keeping with the philosophy behind ERRA, we want a nonparametric approach, rather than one that requires specifying the form of a nonlinear function *a priori* and estimating its parameters using nonlinear regression techniques. As described in Sect. 6 of K2022, ERRA can approximate a catchment's runoff response over multiple time scales, using a piecewise-linear broken-stick function that has closely spaced knots at

short lag times, and widely spaced knots at long lag times (Fig. 16). Each knot represents a weighted average of the runoff response over the range of lag times that is closest to it.

The general approach, as described in detail in K2022, assumes that the runoff response coefficient $\beta_k$ varies linearly as a function of lag time between pairs of knots with (for example) values of $\beta_3^*$ and $\beta_4^*$ at lags of $k = \kappa_3$ and $k = \kappa_4$,

respectively,

$$\beta_k = \beta_3^* \left( \frac{\kappa_4 - k}{\kappa_4 - \kappa_3} \right) + \beta_4^* \left( \frac{k - \kappa_3}{\kappa_4 - \kappa_3} \right) \ . \tag{17}$$

If expressions of this form are substituted into Eq. (3), the linearly weighted regression coefficients between each set of knots can be replaced by weighted averages of precipitation values around each knot $\kappa_\ell$:

$$P_{j,\ell}^* \approx \sum_{k=\kappa_{\ell-1}}^{(\kappa_\ell)-1} \left( \frac{k - \kappa_{\ell-1}}{\kappa_\ell - \kappa_{\ell-1}} \right) P_{j-k} + \sum_{k=\kappa_\ell}^{\kappa_{\ell+1}} \left( \frac{\kappa_{\ell+1} - k}{\kappa_{\ell+1} - \kappa_\ell} \right) P_{j-k} \ , \tag{18}$$





Making this substitution converts Eq. (3) from an equation that estimates coefficients $\beta_k$ for every lag $k$, based on precipitation inputs $P_{j-k}$ at each lag, to an equation that estimates coefficients $\beta_\ell^*$ for knots at lags $\kappa_\ell$, based on weighted averages $P_{j,\ell}^*$ of precipitation at lags surrounding those knots:

$$Q_j = \sum_{\ell=1}^{n_\kappa} \beta_\ell^* \, P_{j,\ell}^* \; + \; \alpha + \varepsilon_j \,. \tag{19}$$

Equations of the form of Eq. (19) can be solved by the same methods used to solve Eq. (3), once the precipitation values

$P_{j-k}$ have been appropriately transformed to the weighted averages $P_{j,\ell}^*$. This implies that any of the methods outlined above for quantifying heterogeneity, nonlinearity, and nonstationarity in runoff response (Sects. 2, 3 and 4, respectively) can also be applied using the multiscale broken-stick approach outlined here. (Readers should note that the methods outlined in Sect. 3 also use piecewise-linear broken-stick models, but for quantifying nonlinear runoff response over specified intervals of precipitation intensity, rather than for quantifying runoff response over wide ranges of lag times. Some of the notation used

above has been recycled from Sect. 3, but these are different broken-stick models used for different purposes.)

### 5.2   Proof of concept: long-tail recession curves at Plynlimon

Here I demonstrate how the methods outlined in Sect. 5.1 can be used to estimate the long tails of recession curves, using hourly data from Plynlimon (Fig. 11) as a proof of concept. Plynlimon nicely illustrates the challenge of estimating

recession behavior, because rain there is so frequent; at Hafren, only 10% of rainless periods are 5 days or longer, and only 2.5% are 10 days or longer. Thus there are few uninterrupted recessions of significant length, so inferring recession behavior on longer time scales will require filtering out the effects of subsequent precipitation inputs, which ERRA is designed to do.

Figures 16a-c show Hafren's weighted average RRD (that is, accounting for nonlinear effects of variations in precipitation

intensity) at all hourly lags up to 1000 hours, or roughly 6 weeks (light blue symbols in Fig. 16a-16c). The RRD is well constrained (as one might expect from over 300,000 hourly measurements spanning 35 years), but plotting on log-log axes (Fig. 16c) reveals that even with such an extensive data set, the long tail of the RRD is not well constrained in percentage terms (note also that several values beyond the range of the axis, including values below zero, are not shown). The amplitude of the noise in the RRD is approximately constant across the full range of lags, and thus is a small fraction of

strong signals (like the RRD peak) but a large fraction of weak signals (like the long tail).

Some of this noise can be suppressed by invoking robust estimation, which is implemented as an option in ERRA via Iteratively Reweighted Least Squares (IRLS; Holland and Welsch, 1977). Robust estimation results from ERRA must be interpreted cautiously because, like any robust estimation method, IRLS downweights points that deviate from the pattern of

behavior exhibited by the bulk of the data, and thus may mistake high-precipitation points as outliers and limit their


influence on the results. This artifact is minimized by applying robust estimation only where, as in Fig. 16, the nonlinear effects of precipitation intensity have been accounted for via the methods of Sect. 3. As the dark blue symbols in Figs. 16a-16c show, robust estimation only slightly reduces the runoff response peak, while having the intended effect of substantially dampening the noise in the recession. Nonetheless the recession remains highly uncertain at long lags.


The uncertainty in the long recession tail is substantially reduced by the broken-stick approach outlined in Eqs. (17)-(19), shown in orange and yellow for non-robust and robust estimation, respectively. In Fig. 16, broken-stick weighted average RRDs are shown for 40 knots that span the full range of lags between 0 and 1000 hours in a nearly geometric progression (an exact geometric progression is not possible because each knot must correspond to an integer lag number). The broken-stick

estimates closely follow the regularly spaced RRD estimates when the signal is strong, and closely follow their average trend when the signal is weak and the individual lag estimates are noisy (Fig. 16c).

Figure 16d presents robust broken-stick weighted average RRDs for seven Plynlimon streams with drainage areas ranging from 0.9 to 10.1 km$^2$ and at least 35 years of hourly streamflow data. As Fig. 16d shows, all seven streams exhibit power-

law recessions that scale as approximately $Q \sim \tau^{-1}$ over lags $\tau$ ranging from roughly 5 hours to roughly 1000 hours. This behavior is markedly different from the exponential recession that would be expected to result from drainage of linear groundwater reservoirs. In particular, if we estimate the recession time scale as $Q/(-\,\mathrm{d}Q/\mathrm{d}\tau)$, we observe that for a general power-law recession $Q \sim \tau^{-\gamma}$, this time scale increases linearly with the lag time itself:

$$\frac{Q}{-\,\mathrm{d}Q/\mathrm{d}\tau} = \frac{\tau^{-\gamma}}{\gamma\,\tau^{-\gamma-1}} = \frac{\tau}{\gamma} \quad . \qquad (20)$$

Such recessions therefore have no fixed characteristic time scale; instead, in our case with $\gamma \approx 1$, the recession time scale at a lag of 10 hours is about 10 hours, the recession time scale at a lag of 100 hours is about 100 hours, and so on. The log-log recession slopes in Fig. 16c are not exactly $\gamma = 1$; instead they range from $\gamma = 1.07$ to $\gamma = 1.21$, deviating from $\gamma = 1$ by between at least 3 standard errors. The corresponding slopes in a conventional recession plot of $\log(-\,\mathrm{d}Q/\mathrm{d}\tau)$ against $\log Q$ (Brutsaert and Nieber, 1977) would be $b = 1 + 1/\gamma$, ranging from 1.83 to 1.93.


The log-log recession slope $\gamma$ can be interpreted in terms of the drainage equation of groundwater storage, if one assumes that streamflow recession, particularly at long lag times, is controlled by groundwater seepage alone and is not significantly influenced by evapotranspiration. Following the analysis outlined in Sect. 6 of Kirchner (2009), a recession that scales as $Q \sim \tau^{-\gamma}$ with $\gamma > 1$ implies a storage-discharge relationship of $Q(S) \sim (S - S_0)^{1/(1-1/\gamma)}$, where $S_0$ indicates the groundwater

storage at which baseflow would go to zero (which would not be reached in finite time). Note that for values of $\gamma$ close to 1, the exponent $1/(1 - 1/\gamma)$ can become quite large (e.g., if $\gamma = 1.2$ the exponent is 6, and if $\gamma = 1.1$ the exponent is 11), implying strong nonlinearity in the storage-discharge relationship. If $\gamma < 1$, a recession that scales as $Q \sim \tau^{-\gamma}$ implies a storage-discharge relationship of $Q(S) \sim (S_0 - S)^{1/(1-1/\gamma)}$, where the exponent is now negative, and $S_0$ indicates a theoretical



upper limit to groundwater storage at which seepage would be infinite. If $\gamma = 1$, seepage becomes an exponential function

of storage, $Q(S) \sim e^{a(S-S_0)}$, where $a$ is the reciprocal of the proportionality constant in $Q \sim \tau^{-1}$ (see Eq. 16 of Kirchner, 2009).

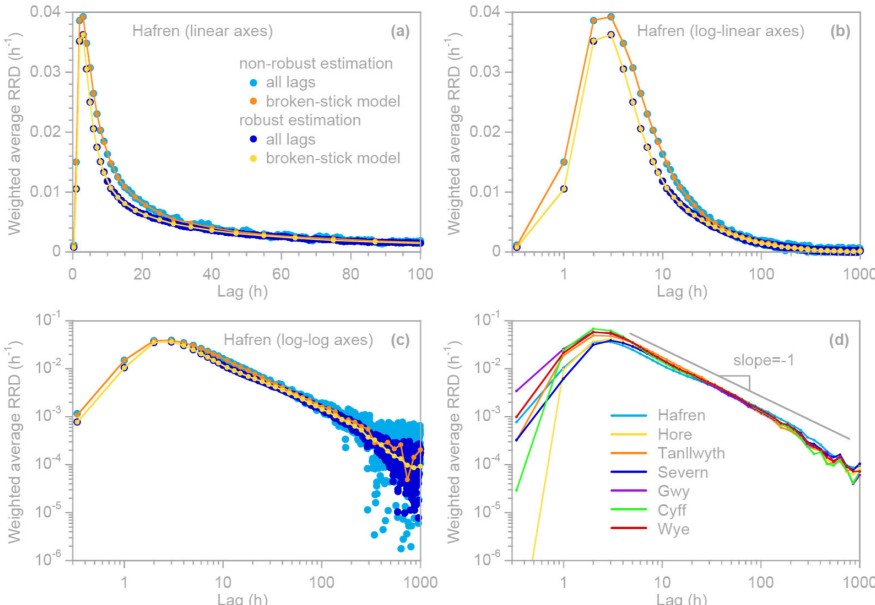

**Figure 16. Long-tail recession behavior at Hafren and six other Plynlimon streams. Panels (a), (b), and (c) show precipitation-**
**weighted average runoff response distributions (RRDs) for Hafren on linear, log-linear, and log-log axes, respectively. Each of**
**these panels shows RRDs calculated at every lag using non-robust and robust estimation (light blue and dark blue, respectively),**
**and calculated using a piecewise-linear broken-stick model over a geometric progression of lag intervals using non-robust and**
**robust estimation (orange and yellow, respectively). At larger lags, the broken-stick approach averages the runoff response over**
**longer lag intervals, thus greatly reducing the scatter in the long tail of the RRD. Robust estimation further reduces the scatter in**
**the RRD tail by limiting the influence of individual data points with large residuals. Panel (d) shows robust broken-stick RRDs for**
**7 Plynlimon streams. All 7 streams have recession limbs that scale as approximately $\tau^{-1}$ over more than two orders of magnitude**
**in lag time $\tau$.**

## 6    Discussion

The proof-of-concept demonstrations in Sects. 2-5 show that ERRA provides a powerful, flexible, and widely applicable

data-driven framework for quantifying how streamflow is coupled to precipitation, including how that coupling varies with

ambient conditions (Sect. 4), with precipitation intensity (Sect. 3), and across the landscape (Sect. 2). These proof-of-

concept demonstrations suggest three broad categories of potential applications. First, ERRA could be applied to quantify



hydrological response for purposes of catchment characterization. ERRA quantifies the coupling between precipitation and
streamflow (rather than just the statistics of streamflow itself). Thus, in inter-catchment comparisons, ERRA should help in
clarifying the effects of differences in landscape characteristics (e.g., soil depth and bedrock lithology), by factoring out
potentially confounding site-to-site variability in precipitation patterns. Second, both in site-to-site comparisons and in
longitudinal studies at individual sites, ERRA could be applied to quantify how changes in factors like climate, land cover,
and land use have altered the coupling between precipitation and streamflow. ERRA could, for example, be applied to
distinguish cases where streamflow patterns have shifted because precipitation patterns have shifted, and cases where the
relationship between precipitation and streamflow has changed. Third, ERRA could be applied to define signatures of
hydrological response for comparison with models. Figures 8, 9, 13, 15, and 16 are all "fingerprints" of hydrological
behavior that could be quantified from real-world data, and then from model simulations, to facilitate model-data
comparisons. Such targeted approaches for confronting models with data are likely to have greater diagnostic power than
goodness-of-fit statistics applied to hydrological time series (Kirchner et al., 1996; Kirchner, 2006).

The illustrative examples presented here have focused on the coupling between precipitation and streamflow, but the
underlying mathematical methods are general and could potentially be used to quantify linkages between many other
hydrological inputs and outputs. For example, ERRA could be used to explore the coupling between precipitation and
groundwater recharge, wherever groundwater level time series allow recharge rates to be estimated from rates of water table
rise. Such an approach would help in characterizing how the vadose zone modulates recharge response to precipitation
inputs. Alternatively, groundwater recharge could be considered as the input and streamflow as the output, to characterize
how the saturated zone mediates streamflow response to recharge fluxes. Similarly, changes in soil moisture could
potentially be used as measures of infiltration rates, and ERRA could potentially be used to explore their coupling to
precipitation patterns and recharge dynamics. Or eddy-flux data or sapflow time series could potentially be used within
ERRA to explore how evaporation and evapotranspiration respond to precipitation inputs, while accounting for vapor
pressure deficit and available energy as co-variates. Beyond a purely hydrological focus, runoff response distributions
quantified by ERRA could also be combined with transit time distributions quantified by ensemble hydrograph separation
(Kirchner, 2019; Kirchner and Knapp, 2020) to estimate the "forward" transit time distribution and explore how it varies
with ambient conditions and precipitation intensity. And ERRA could be used to explore the coupling between precipitation
time series and chemical fluxes in streamflow, to quantify how solute fluxes respond to variations in precipitation forcing
and antecedent conditions. Several of these potential applications are currently under investigation.

Although the proof-of-concept demonstrations presented here have been based on hourly data, the mathematics of ERRA do
not tie it to any particular time scale, and it has been tested with everything from 10-minute data to daily data. As shown in
Figs. 8-10, changes in the time resolution of the underlying data can reveal some features and conceal others. In general, if
the frequency of the underlying data is too high compared to the timescale of runoff response, the standard errors of the RRD





and NRF will be large because the residuals will be strongly autocorrelated, reflecting the difficulty in distinguishing runoff responses at closely spaced lags. Conversely, if the frequency of the underlying data is too low compared to the width of the

runoff response peaks, they will be damped by averaging. Furthermore, in low-frequency data, it may become difficult to distinguish between nonlinear response to variations in precipitation intensity and nonstationary response to variations in antecedent wetness, because (for example) precipitation falling early in the day will contribute to the wetness of the landscape, and thus to greater runoff response, later in the day. In this example, runoff response to daily averaged precipitation will inherently combine the direct effects of precipitation intensity (in relation to infiltration capacity, for

example) and the effects of precipitation on catchment wetness during the same time step.

At the risk of stating the obvious, results from ERRA (or indeed from any data-driven technique) will only be as good as the data that are available. While ERRA can do a lot to filter out confounding factors (particularly where the potential confounders have themselves been measured), it will be inherently vulnerable to artifacts in the underlying data. Users are

therefore strongly encouraged to visually – not just statistically – inspect their data for problems rather than blindly applying ERRA or other analysis tools. Users are also encouraged to avoid sub-dividing the input data into too many categories or time periods, leaving ERRA too little information to work with (although this should be evident in the standard errors, which ERRA reports by default for all of its results). Because ERRA is based on temporal correlations between inputs and outputs, it will likely struggle to identify linkages where those correlations are weak. For example, whereas the temporal correlations

between rainfall and streamflow are relatively strong, the temporal correlations between snowfall and streamflow are much weaker, because the timing of snowmelt is highly variable and controlled primarily by energy fluxes to the land surface rather than by precipitation per se. ERRA will not work well, and indeed should not work well, where major variations in streamflow are controlled by drivers that are not accounted for in the input data (e.g. snow or glacier melt, or dam releases).

Users should also take care to ensure that the time series that they use are measured on consistent time bases. This can be more challenging than it might seem, because weather and streamflow data may be provided by different agencies using time stamps based on different time zones, and sometimes with shifts between summer and winter time. Even daily data sets may be based on different definitions of when a "day" begins and ends: at midnight UTC, at midnight local time, or at some other hour (which may again shift between summer and winter time). A further difficulty is that these important details are often

poorly documented, but can significantly affect the results of ERRA analyses.

It should be clear that ERRA is not a simulation model in the conventional sense. The goal of ERRA is analysis and characterization rather than prediction, because RRDs and NRFs are at best incomplete descriptions of hydrological behavior, even when the nonlinearity, nonstationarity, and spatial heterogeneity in that behavior are accounted for. RRDs

and NRFs are aggregated descriptions of behavior, averaged over ensembles of events. Thus one should not expect them to yield nice goodness-of-fit statistics if they are used for hydrograph prediction. That is not their purpose.



It should likewise be clear that ERRA is designed as a tool for iterative, hands-on exploration of hydrological data, through trial and error with analyses of varying degrees of complexity. Thus although ERRA is computationally efficient and could be blindly applied to the massive hydrological data sets that are now becoming available, its primary intended purpose is not data mining per se. It is designed for human learning rather than machine learning.

Last but not least: analyses like those presented here should be the beginning, not the end, of a scientific investigation. They characterize how hydrological systems behave, but do not, at least by themselves, explain why. Answering "why" questions will require carefully designed hypothesis tests, including those that are encoded in models. But here too, ERRA can play a role, helping to test alternative models of "why", by comparing their signatures to the signatures of real-world behavior.

**Code and data availability**

The ERRA script, introductory documentation for users, and scripts and source data for all of the figures in this paper will be posted to a durable public archive with a permanent doi after final acceptance.

**Acknowledgments**

I thank the US Geological Survey and the MOPEX study for making available the data analyzed in Sects. 2 and 3, and the Centre for Ecology and Hydrology and the Plynlimon field staff for making available the data analyzed in Sects. 4 and 5. I thank Paolo Benettin, Wouter Berghuijs, Harsh Beria, Cansu Culha, Zahra Eslami, Xiahong Feng, Marius Floriancic, Huibin Gao, Minhui Li, Shaozhen Liu, Ilja van Meerveld, Cristina Prieto, Hansjörg Seybold, Zhuoyi Tu, Jian Wang, Tetiana Zabolotnia, and Maria Grazia Zanoni for helpful discussions and particularly for beta-testing the ERRA code and documentation.

Permission for re-use/modification of Fig. 1:
https://www.annualreviews.org/page/authors/author-instructions/distributing/permissions

Permission for re-use/modification of Fig. 11:
https://www.agu.org/publish-with-agu/publish/agu-publications-policies

**Competing interests**

The author has declared that there are no competing interests.





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
