# Peer review of "Characterizing nonlinear, nonstationary, and heterogeneous hydrologic behavior using Ensemble Rainfall-Runoff Analysis (ERRA): proof of concept"

_Hydrology and Earth System Sciences, 2024_

## Author Response (AR1)

**I thank the editor and reviewers for their comments. Here I reproduce their comments (in plain text), with my responses in bold. Details concerning what has been changed are bold and underlined.**

Comments from editor (Thom Bogaard):

Dear dr. Kirchner, thanks for the submission of the ERRA work to Hess. You received two in-depth reviews for which I thank the referees. They invested significant time in getting to the bottom of ERRA approach. Both referees agree it is interesting and well described research. The paper fits well in Hess. The paper is elaborative in recognizable style: detailed and rigorous. The paper is an additional loot to the analytical hydrological models and is, as you write, intended for human learning over machine learning. A very welcome contribution in the ML-fitting epoch we are experiencing.

**Thank you.**

However, there are a few points both the referees bring up that could improve the readability and impact of the work. First of all, the paper should be shortened especially in section 1.1. The exhaustive description of travel times is not directly of importance for ERRA, but contextual. A few lines of text, references and figure 1 would be sufficient, I think.

**I have shortened section 1.1 by roughly 50 lines of text. Section 1.1 is now about 9 percent of the length of the manuscript, and the discussion of travel times is only about 15 lines (depending on exactly how one chooses to count). The travel-time context is important because, in view of the extensive literature on travel times in recent decades, readers need to understand that response times are something quite different. This is doubly important because in much of the unit hydrograph literature, response times and travel times are assumed to be the same thing.**

Second, I am fine with the modest overlap with K2022, it makes sense to repeat some of that in this paper. No changes required here.

**Thanks. I obviously agree.**

Third, the discussion on uncertainty and identifiability of parameters. In the paper it is mentioned there are uncertainty analyses, which are not reported in the paper (and as such should then also not be mentioned in abstract and intro). I am OK not adding an extensive uncertainty analysis in the main text, the paper is convincing as it is. If indeed at some point there will be a formal (like you state 'straightforward') uncertainty analysis provided, of course, this can be mentioned (e.g. at data availability or supplement or the like). Moreover, you did agree on adding some more info on parameter uncertainty and identifiability (number of subcatchments, too long lag-times or weak signals), this will indeed be good to discuss (again, without an extensive analysis in the main text, maybe in supplement or in the data provided).

**This is now addressed in a new Section 6, titled "Limitations". Just to be clear: An uncertainty analysis does not need to be "provided", because ERRA performs uncertainty analysis by default, and reports standard error estimates for all of its results. I have now added explicit statements to Sections 2.1 and 6 to point this out to readers (particularly lines 204-208, 938-945, and 978-979).**

Finally, the paper ends with a discussion section which is a combination of discussions, limitations and outlook. I suggest to have two sections: one on discussion of applicability and limitations of ERRA as you have shown with the proof-of-concepts, including the practical aspect of the minimal data set size you foresee is required, the equifinality issue when moving to too large number of subcatchments and too large signal dampening and the like. Then finalizing the paper with conclusions and outlook of all papers in progress (L941-L957). I look forward seeing them out.

**I appreciate this suggestion. There is now a new Section 6, titled "Limitations", and a new Section 7, titled "Applications and outlook". These two sections include the content that was formerly in Section 6 ("Discussion"), with an additional overview of data uncertainties and limitations.**

I look forward receiving the revised version which will be evaluated by me.

**Thank you. Note that the code and documentation will be available at the stated doi (which has been minted and reserved) after final publication of the paper. This is necessary so that the doi of the paper can appear in the code and documentation, at the same time that the doi of the code and documentation can appear in the paper.**

Comments from Reviewer #1:

This manuscript is a welcome addition to a series of analytical hydrological models. As the examples illustrate nicely, the idea to linearize inherent nonlinear responses opens up statistical methods that will give the hydrological community an entirely new set of tools to explore for some time into the future. This is a landmark paper.

**Thank you.**

Although I found no errors in the manuscript, I found opportunities to improve clarity, with three general comments and several minor ones.

1. The discussion gives useful context, but an important unaddressed issue is the highly parameterized nature of ERRA. I would have liked to read some discussion of how important this is in terms of model overfitting and uncertainty, and in terms of interpretation of parameters. The VPD example brings this issue into sharp focus.

**Because ERRA is essentially just solving very large linear regression problems, surrounded by various transformations of variables, the math and statistics of the uncertainty analysis are straightforward (and are explained in Kirchner 2022). Model overfitting should be obvious in the inflation of the standard errors, which ERRA provides for all its results (and which are shown in the results plotted in the manuscript).**

**Issues of parameter uncertainty and overfitting are now addressed in a new Section 6, titled "Limitations". ERRA performs uncertainty analysis by default, and reports standard error estimates for all of its results. I have now added explicit statements to sections 2.1 and 6 to point this out to readers (particularly lines 204-208, 938-945, and 978-979).**

2. In several places, ERRA is described as time-dependent, but it only inherits variations in time from variation in precipitation (sec 2-3) or watershed state (sec 4). The P-k-NRF relationships themselves

do not vary through time because Eqs 11, 16, and 19 treat time variability as error. Thus, ERRA is lag-dependent but not time-dependent. It's a fine point, but clearly separating inherent dependence from inherited dependence would improve clarity.

**One can also make ERRA results explicitly dependent on time itself, for example by separating the time series into different decades, different seasons of the year, daytime vs. night-time, etc. (instead of, for example, wet vs. dry conditions). In the manuscript I tried to illustrate some of the things that ERRA can do, but made no attempt to be comprehensive, because the paper is already rather long. However, I now mention that users can choose splitting criteria that make ERRA explicitly time-dependent (lines 635-637).**

3. Some organizational choices made the paper difficult for me to follow. The Roanoke River example is given before the theory necessary to support it, which made me wonder why I didn't understand where the results were coming from. Also, I was puzzled by Fig 6b until I got to the text, three sections later, that explained the utility of the avg RRD.

**I assume that by "the Roanoke River example", the reviewer is referring to Figure 4. Figures 4a and 4b are based on Section 2.1, which comes before them. Figure 4c is based on Section 2.3, which follows it. The only way to avoid this would be to split Figure 4 into two figures, which would add yet one more figure, and would make it difficult for readers to compare the three panels (which is one of the main points of Figure 4 in the first place). I have added signposts in the caption to Figure 4 to alert readers to the sections each panel refers to (lines 258-261).**

**The story with Figure 6b is similar. In principle one could split it in half and show Figure 6a in Section 3.1 and Figure 6b in Section 3.4 (which is 3 sub-sections later, not 3 sections later). But doing that would add yet another figure to a paper that already has 16 of them, and would split two panels that conceptually belong together. Instead I have added signposts to the figure caption to tell readers where they can find the necessary details (lines 412 and 416).**

Detailed comments:

L205 this passage implies that ERRA is not based on linear superposition even though it is. It's not the linear superposition that is the difference between ERRA and unit hydrograph, it's the nature of the functions being linearly superposed.

**The passage in question is "Unit hydrograph methods are also based on the premise of linear superposition, such that the response to x units of rain falling at time t is assumed to be x times the response to a single unit of rainfall at time t. By contrast, ERRA recognizes that streamflow may respond nonlinearly to variations in rainfall intensity, and uses nonlinear deconvolution methods to quantify that response"**

**Unit hydrograph methods assume that the effects of 2, 3, or 4 units of rain falling at time t must be 2, 3, or 4 times as large as the effects of 1 unit of rain at time t. That is linear superposition. ERRA can, at the user's option, replace the linear assumption of a unit hydrograph with nonlinear functions of precipitation intensity that are determined by the data and, importantly, that vary with lag time. This is fundamentally different from stuffing the precipitation time series through a**

**nonlinear transform and then putting the results of that through a unit-hydrograph type of deconvolution.**

**One could perhaps say that this is linear superposition of nonlinear functions of precipitation intensity that vary with lag time... but then that's not linear superposition _of precipitation inputs_. The relationship between inputs and outputs is no longer linear (or even a simple nonlinear transformation).**

Similarly, the passage L207-8 implies ERRA is nonstationary even though it is stationary. The apparent nonstationarity is because ERRA can incorporate time-varying inputs to generate nonlinear responses, but there is no time dependence in the model itself. Again, these are fine points, but I found this paragraph made it difficult for me to understand ERRA for the first time because it led me to expect non-additive, time-dependent structure.

**The passage in question is "Furthermore, conventional unit hydrograph methods assume that runoff response is stationary, such that a given unit of rain always has the same effects on runoff, regardless of when it falls. By contrast, ERRA combines deconvolution and demixing methods to explicitly quantify how runoff responses vary with ambient conditions, even if those runoff responses are overprinted on one another"**

**The passage explains what it means by "stationary", and explains how ERRA is different. I will see if there is a clearer way to say this, but what is already there was the product of considerable thought and care.**

**And there can indeed be "time dependence in the model itself", as outlined in the response to the reviewer's point 2 above. I have modified the passage quoted above to say "how runoff responses vary over time or vary with ambient conditions..." (lines 164-165)**

L259 if I understand correctly, beta is only reduced by half if rainfall is constant during the time step and the response is linear. Using the 0.5 simplification should produce useful results if timesteps are short relative to characteristic timescales of the RRD and rainfall variations, but full analytical rigor requires resolving variations within the interval. Isn't this in fact the source of timescale variations discussed in section 3.3?

**"Resolving variations within the interval" is impossible in practice, because rainfall data are inherently integrated over the sampling time step. The factor of 0.5 arises, not from an assumption that rainfall is constant, but instead as an average over stochastically fluctuating rainfall rates, and over all the time steps in the regression analysis. I have rewritten the passage in question to make clear that the factor of 0.5 is not assuming constant precipitation, but instead averaging over stochastic precipitation (lines 217-218).**

**Yes, the stochasticity of rainfall inputs is the main source of the timescale variations shown in Section 3.3 (as is already pointed out there, for example in the first sentence: " The stochastic nature of precipitation means that precipitation rates measured over different time intervals will have different distributions"). More precisely, it is the timescales of catchment response (partly reflecting the stochasticity of rainfall) that can lead to the need for aggregation of time steps, to avoid excessive serial correlation in residuals.**

L458 I think this was done using linear regression to obtain parameters of Eq 11, correct? It would be helpful to specify.

**Correct, and good catch. I have added "as estimated by Eq. 11" (lines 425-426).**

L467 it would be more correct and helpful to term this "… within a range of rainfall intensities" (as L543).

**Not exactly. The NRF values are functions like those shown in Figure 6a, evaluated either at the knot points, or at the weighted averages of precipitation within each segment. That is, the NRFs correspond to specific precipitation rates, not ranges. But to avoid confusion, I have added "or over a given intensity range" to L435.**

L522 what are we to make of the peak lags that vary by up to 100% depending on timescale of aggregation for low-intensity events? I would assume this is because low-intensity, short-duration events have less-identifiable peaks, but the standard errors of the estimated mean lags are small. Can we interpret this aggregation-level dependence in terms of runoff generation mechanisms?

**I would be cautious about interpreting this as more than a broad pattern of variation, for two reasons. First, as noted by the reviewer, low-intensity events often have indistinct peaks, as one can see from Figure 8. Second, when precipitation is averaged over longer time steps, one can resolve lower (average) precipitation rates more precisely, but at the cost of resolving the timing of runoff response less precisely.**

L681 it is probably worth mentioning that using antecedent Q probably becomes worse as a proxy for watershed wetness as the size of the watershed increases.

**Good point (but with emphasis on "probably"). I have added "(this may be a less effective proxy in large catchments with long lag times)" to the passage in question (lines 662-663).**

Fig 12 I understand the negative values of the RRD/NRF at short lags to be due to recession prior to arrival of new peaks, but how can we interpret negative values at longer lags? If they are also due to recession, I would think negative values would occur for low-intensity rain and high antecedent Q, but here they occur when antecedent Q is low.

**I would primarily interpret these as artifacts of jointly estimating coefficients for very strong signals (responses to high precipitation rates and antecedent moisture) and very weak signals (responses to low precipitation rates and low antecedent moisture). It is inherently difficult to accurately estimate weak signals in the presence of strong signals; it's a bit like trying to hear whispers at a heavy metal concert. ERRA does a very good job of suppressing statistical cross-talk between the coefficients of the strong signals and the weak ones, but it isn't perfect. Basically, if the least-squares algorithm can get a slightly better fit by driving the low-antecedent-Q coefficients negative, it will ruthlessly do it. Streamflow is largely insensitive to low-intensity precipitation and low-antecedent-wetness conditions, particularly at long lags, so it is difficult to estimate the corresponding coefficients accurately.**

**I have added the following paragraph (lines 704-722): Readers will notice that several of the RRD and NRF values in Fig. 12 are below zero. ERRA does not artificially constrain NRF and RRD coefficients to be non-negative, so small negative values may occur from time to time for at least three reasons. The first is random statistical fluctuations: if the true value of a coefficient is positive but small, random noise may lead to stochastic fluctuations in the coefficient estimates that occasionally dip below zero. In this case, one would usually expect the error bars to be roughly as large as the deviation below zero. This is not the case in Fig. 12, where the error bars are almost always smaller than the plotting symbols. A second reason for negative values can be residual autocorrelation that is too strong to be adequately compensated by the ARMA noise**

**correction procedure, which can lead to spurious patterns in the NRF or RRD coefficients, and to underestimation of the error bars (for this reason, ERRA issues warnings when it detects strong residual autocorrelation). A third reason can be confounding variables that are not included in the analysis, and whose effects are aliased as distortions of the NRF or RRD coefficients. All three of these phenomena can be amplified when – as here – one tries to jointly estimate coefficients for very strong signals (from high precipitation rates and wet antecedent conditions) and very weak signals (from low precipitation rates or dry antecedent conditions). Small variations in the coefficients of a strong signal can sometimes be offset by larger variations in the coefficients of a weak signal, and least-squares fitting will make such a tradeoff if it leads to a closer match to the observed streamflow. ERRA does its best to suppress statistical cross-talk between the coefficients of the strong signals and the weak ones, but this is an inherently difficult task. Because streamflow is relatively insensitive to low-intensity precipitation and rain that falls under dry conditions, it is difficult to estimate the corresponding coefficients accurately, particularly at long lags. Nevertheless, Fig. 12 clearly shows the main peak of the runoff response to both high-intensity and low-intensity precipitation falling under both wet and dry ambient conditions.**

Comments from Reviewer #2

The paper proposes a series of variants of a data driven model (ERRA) for interpreting / describing the relationship between rainfall and runoff. The model proposed is supported by a series of applications to real-world case studies, aimed at showing the potential of the tool. The main idea is that of using the total rainfall instead of the effective rainfall in establishing a relationship between rainfall and runoff, though keeping the original convolutive structure of the standard IUH approach.

**Using total rainfall rather than effective rainfall is not, by itself, "the main idea". As outlined in Section 1.2, ERRA differs from the standard IUH approach in multiple ways (of which 4-7 in the following list are most important):**

**1) It uses total precipitation rather than "effective' precipitation (as pointed out by the reviewer).**

**2) It uses total streamflow rather than "storm runoff" and thus does not involve making additional assumptions for baseflow separation.**

**3) It does not require defining and isolating individual events, but instead extracts information from the entire streamflow and precipitation time series.**

**4) It does not necessarily assume that runoff response scales linearly with precipitation intensity. Instead it can quantify, from data, nonlinear relationships between precipitation intensity and runoff response.**

**5) It does not necessarily assume that runoff response is always the same. Instead it can quantify, from data, how runoff response varies with ambient conditions (such as catchment wetness, which in turn depends on antecedent precipitation).**

**6) It can explicitly quantify runoff responses from different parts of a catchment, even if those subcatchments are not individually gauged and instead their runoff is combined at the catchment outlet.**

**7) Its goal is not to predict hydrographs, but instead to analyze the relationship between rainfall and runoff, and how that relationship varies with precipitation intensity and ambient conditions.**

This idea has been already presented in another paper, but in this contribution the same idea is expanded and applied to a set of case studies.

**The other paper (Kirchner 2022, hereafter K2022) presents a general mathematical framework for estimating nonlinear, nonstationary impulse response functions in heterogeneous systems.  It is a signal processing paper.  It has nothing in particular to do with rainfall-runoff relationships or unit hydrographs per se.**

The paper has been written with care and is generally clear. That siad, I see several points that need careful consideration in terms of language, organization, assumptions, presentation, internal consistency.

**Thank you.**

The major issues I see are summarized as follows:

- The introductory part is way too long, and the intial detour about the difference between travel time distributions and response functions is not relevant to this work. The paper is already very long, and travel times are not a main issue of this paper. Likewise, Figure 1 is not necessary. Please eliminate all the information that is not functional to understand the novel contribution of this paper;

**The introduction is about 12 percent of the manuscript.  This can be shortened somewhat but this will not dramatically shorten the paper.  (I know the paper is long, but it covers a lot of material.)**

**I agree that travel time distributions are not the focus of this work, and this can be shortened and de-emphasized in the revised manuscript.  However, the literature of recent decades has been so saturated with travel time studies that the distinction between travel times and response times needs to be emphasized.**

**The contrast between travel times and response times also needs to be discussed because in much of the IUH literature, response times and travel times are assumed to be the same thing.**

**Figure 1 is necessary for two reasons.  First, because it clearly illustrates the difference between response times and travel times.  And second, and more importantly, because it explains what I mean by runoff response, which is not discharge itself, but the incremental change in discharge due to a precipitation input.**

**I have shortened section 1.1 by roughly 50 lines of text.  Section 1.1 is now about 9 percent of the length of the manuscript, and the discussion of travel times is only about 15 lines (depending on exactly how one chooses to count).**

- There is some overlapping with existing work by the same author (which is referred to as K2022 in the paper) and the paper should pinpoint more clearly what is novel in this paper as compared to the existing literature. It seems to me that there are about 250-300 lines before the first novel contribution is presnted. I suggest that previous work should be condensed, and the novel contribution of this Ms. should be better emphsized

**The previous paper (Kirchner 2022) is the mathematical foundation of the present manuscript. It presents the mathematics for estimating impulse response functions in nonlinear, nonstationary, and heterogeneous systems. There is essentially no overlap in goals or scope, except of course that the previous paper provides the mathematical fundamentals underlying the present one.**

**The previous work is long and technical, running 44 pages in print, and all that is presented here are several key equations from that paper, translated into terms that are familiar in hydrology, so that readers can understand the basics of how ERRA works. Thus the previous work has already been massively condensed in the present manuscript.**

- Spatial heterogeneity is not resolved but somehow arbitrarily included in the model in a parametric form assuming that the role of rainfall heterogeneity can be described superimposing the contribution to the discharge of two or more point rainfall records (each of which is coupled with a different subset of response functions). What I'm trying to say is that, while the procepdure proposed by the author is reasonable, the terms $f_a$ and $f_b$ in eq 4 are very likely dependent on factors not included in the analysis, such as the spatial distribution of soil moisture, the intensity of rainfall in different regions of the catchment, and the vectors $P_A$ and $P_B$ themselves.

**This is not accurate. As stated in the manuscript, the constants $f_a$ and $f_b$ represent the fractions of the catchment area that can be considered to receive one or another precipitation time series. (A similar partitioning would also need to be done by any simulation model driven by rain gauge data.)**

**The constants $f_a$ and $f_b$ must be specified a priori; they are not free parameters, because they rescale the response distributions and therefore it is mathematically impossible to estimate them from the same data. The factors mentioned by the reviewer will influence the response distributions, but not the values of $f_a$ and $f_b$.**

- Moreover, spatial heterogeneity can be described only at the cost of increasing a lot the number of parameters: for instance, if the available rainfall stations in the catchment are 10 the number of params would be ten times larger as compared to the equivalent homogeneous case; if the response function associated to the different parts of the catchment can not be "trained" by discharge data (beacause of the lack of streamflow data in correspondence of intermediate gauging stations) the chance of getting over-parametrized is very high.

**The manuscript illustrates how the method works for two stations. It does not claim that it works for an arbitrarily large number of them. The limitation would be statistical rather than mathematical: if the rainfall records are too similar to one another, or if the runoff response is sufficiently damped that it does not reflect the individual signals coming from the separate subcatchments, then it may not be possible to distinguish the runoff response from different parts of the basin. But if this is the case, it will be obvious from the standard errors of the estimated response distributions. I have added the qualifier "(as long as the subcatchments are not too numerous, or their precipitation records are not too similar, which would be reflected in very large standard errors in the reported RRDs)" near the end of Section 2 (lines 340-342). I have also added the qualifier "as long as their time series are sufficiently distinct" to line 306.**

- In the light of the empirical nature of the procedure designed to capture the effect of rainfall heterogeneity I think that the emphasis on the ability of ERRA "to quantify heterogeneous runoff response in different parts of the catchment, even if those subcatchments are not separately gauged" is not justified (this ability can not be proved in the absence of measures in different

subcatchments, and this has not been done in the paper by formally comparing different models accounting for the numebr of parameters)

**It's not quite correct that ERRA is "designed to capture the effect of rainfall heterogeneity". Instead, it _exploits_ rainfall heterogeneity to quantify how different parts of a landscape _respond_ to rainfall (even if those subcatchments are not separately gauged).**

**In any case, the reviewer is correct: the manuscript presents a demonstration of this approach, but not a field test of the method against subcatchment gauging data.  More general benchmark tests against synthetic data have already been performed in K2022.  To make this clear, after Eq. 5 I have added, "The approach outlined in Eqs. (4)-(5) has been benchmark-tested using synthetic data in Sect. 3 of K2023; here I show a simple application, for purposes of demonstration, using the Roanoke River basin" (lines 308-310).  I hope to present a more detailed field test against subcatchment gauging data in the future, but realistically that would be a whole paper in itself.**

- the linearity is not fully eliminated, as the proposed formulation retains the convolution structure of the standard IUH, and the superposition principle is used throughout the paper (the superposition principle is still valid for non-overlapping inputs, since the introduced non-linearity is not a full non-linearity). The author is clearly well aware of this, and the use of the term "non-linearity" in a system which is somehow linear but in which the response is dependent on the input is questionable, though being technically acceptable and correct.

**As the manuscript points out in Section 3.5, the distinction that we need to make is between systems in which runoff is a nonlinear function of a modeled catchment storage (which may be what the reviewer means by "fully nonlinear" below) and those where the incremental runoff response is a nonlinear function of precipitation intensity (which is what in meant by "nonlinear" in the context of ERRA).**

- However, there are several statements in the abstract an in the paper that might be misleading as they could suggest that this model is non-linear in a broad sense (meaning that there is no response function, and the superposition principle does not apply). The readers should be made aware of the fact that the formulation assumes linearity in the model structure, and that the superposition of the effects still applies - as in the original IUH theory (at least for non-overlapping inputs).

**I agree that some terminology (such as "nonlinear system") could be confusing here, and I have now made sure that in all cases, I refer to nonlinear _behavior_ or nonlinear _runoff response_ (except in the case of Eq. 13, which is actually a nonlinear system).  What ERRA does is characterize the nonlinear dependence of runoff behavior on precipitation intensity.  In a system where Q is a lagged function of (say) P^2, so that twice as much precipitation doesn't generate twice as much streamflow, but instead four times as much, the principle of superposition will not rigorously apply, but ERRA can still estimate how lagged Q varies as a function of P (as illustrated in Section 3, and benchmark tested in K2022).  And as described below, ERRA can estimate ensemble average runoff responses even when the system is non-linear in the broad sense even though the superposition principle does not strictly apply.**

- The argument that the response fucntions are dependent on the antecendet wetness (or the antecednt Q, as proposed in the application) creates a paradox. If this is the case – if the response function depends on the catchment wetness - than it is pretty clear that the system is fully non-linear and the response function NRF is not dependent only on the rainfall input that has

generated such response, but actually depends on the entire rainfall time series, which contradicts the basic hypothesis of the first part of the proposed work (the possibility to define a response function and the use of the superposition principle for non-overlapping inputs).

**It is important to recognize that the RRD and NRF are estimates of the ensemble average response of the system to rainfall inputs, averaged over ensembles of many precipitation events. The reviewer is correct that in a nonlinear system, responses to individual precipitation inputs will differ, depending on both past precipitation (which determines antecedent wetness) and future precipitation (which spurs the release of additional water from storage). But the goal of ERRA is not to analyze or predict runoff response to individual precipitation events, but instead to characterize the general pattern of runoff response, averaged over many events.**

**Figure 1e illustrates this distinction. As Figure 1e shows, the light blue lines depicting the individual runoff responses vary substantially, but the dark blue line – the ensemble average runoff response distribution – can be defined with increasing accuracy as more and more events are sampled. In other words, although the superposition principle does not apply at individual points in time, it is accurate enough, averaged over large ensembles of points, that the average response can be characterized.**

**In a nonlinear system, the average runoff response distribution will not accurately predict the hydrograph, but that is not ERRA's purpose. Its purpose is to characterize the average response. I have verified that it does this accurately, even in nonlinear systems, using benchmark tests with nonlinear bucket models, but that work will need to appear in a future paper because it is too long and technical to include here.**

- The model is claimed to be non-parametric but there are actualy lots of parameters (all the beta values and many others). The identification of such parameters is a serious issue here, but is not discussed in the paper

**In data analysis and statistics, terms often have multiple meanings, depending on context. However, in any of these contexts, "non-parametric" does not mean "having no parameters".**

**In statistics, "non-parametric" means a technique is distribution-free, meaning that it does not assume that the sample comes from a particular parametric family of probability distributions. More generally, "non-parametric" means that one does not need to assume a functional form and estimate parameters for it.**

**In estimating the impulse response of a system, an approach that estimates the parameters of a gamma function is called "parametric", and an approach that estimates a series of lag coefficients – like ERRA does – is called "non-parametric". But OK, I have removed "nonparametric" from the abstract. In the other two places the term is used, it should be clear from context what I am talking about.**

**On the more substantive issue here, yes, users could potentially try to estimate too many coefficients from time series that are too short or too indistinct, leading to identifiability problems. But any such problems will be obvious from the standard errors that ERRA reports by default with all of its results (and that are shown in the results plotted in the manuscript). Because ERRA analyses are, at their core, linear regressions (albeit often complicated ones), the calculation of standard errors in ERRA follows straightforward regression statistics and first-order, second-moment error propagation.**

**Section 6 of the revised version contains a more detailed discussion of parameter identifiability and uncertainty.**

- Uncertainty analysis is mentioned in the abstract but is not discussed in depth in the paper as far as I can see (and thus should be removed from the abstract). The author is encouraged to discuss the minimum data requirements for the application of this type of models (and which is the typical uncertainty in the estimation of the model params).

**The sentence in question is, "An R script is provided to perform the necessary calculations, including uncertainty analysis."  This sentence announces the availability of the R script, and outlines what it does.**

**As indicated above, the math and statistics of the uncertainty analysis are straightforward (because this is ultimately just least-squares regression, surrounded by various transformations of variables), but I can include an overview of this in the revised version of the manuscript.**

**The minimum data requirements will depend on many factors, including the quality of the measurements, the complexity of the analysis that one wants to conduct, the influence of confounding factors, the characteristics of the rainfall forcing (how frequent are precipitation events?), and the characteristics of the system itself (in particular, how damped its response is). So it's not possible to provide general guidance about minimum data requirements.  If the data are insufficient, the standard errors will show it.  This is discussed in Section 6 of the revised manuscript.**

A copule of additional in line comments:

Explicit references to the ability of ERRA to filter out the effects of overlapping of different impulse responses and to the possibility to interpret NRF as the response to an impulsive rainfall event can be found in lines: 28-29, 412-413, 539, 542, 821-822, 862. It should be pointed out that this is possible only because of the particular type of (weak) non-linearity of the proposed model (see point 3 above)

**The NRF is not "the response to an (individual) impulsive rainfall event".  It is an estimate of the ensemble average of the responses to many rainfall events.  As noted above, this does not require weak non-linearity and instead can function reliably even in systems with broad-sense nonlinearity.**

Figure 12 shows some RRD and NRF having negative values for some lags, which is highly confusing - a detailed explanation is needed

**This is an example of a common phenomenon in statistical analyses of data sets that mix strong and weak signals: small errors in estimating the strong signals can be paired with larger (in percentage terms) errors in the weak signals.**

**ERRA does not artificially constrain the coefficients to be non-negative, so small negative values may occur from time to time for at least three reasons.  One is random statistical fluctuations; in this case the error bars will typically be roughly as large as the deviation below zero.  A second reason is residual autocorrelation that is too strong to be adequately compensated by the ARMA noise correction procedure; ERRA will try to warn about this, because it can lead to underestimation of the error bars.  A third reason can arise when  – unsurprisingly – the real-world**

system deviates from the statistical model, for example due to confounding variables that are not accounted for.

After Figure 12, the revised manuscript now includes a discussion of these issues: " Readers will notice that several of the RRD and NRF values in Fig. 12 are below zero.  ERRA does not artificially constrain NRF and RRD coefficients to be non-negative, so small negative values may occur from time to time for at least three reasons.  The first is random statistical fluctuations: if the true value of a coefficient is positive but small, random noise may lead to stochastic fluctuations in the coefficient estimates that occasionally dip below zero.  In this case, one would usually expect the error bars to be roughly as large as the deviation below zero.  This is not the case in Fig. 12, where the error bars are almost always smaller than the plotting symbols.  A second reason for negative values can be residual autocorrelation that is too strong to be adequately compensated by the ARMA noise correction procedure, which can lead to spurious patterns in the NRF or RRD coefficients, and to underestimation of the error bars (for this reason, ERRA issues warnings when it detects strong residual autocorrelation).  A third reason can be confounding variables that are not included in the analysis, and whose effects are aliased as distortions of the NRF or RRD coefficients.  All three of these phenomena can be amplified when – as here – one tries to jointly estimate coefficients for very strong signals (from high precipitation rates and wet antecedent conditions) and very weak signals (from low precipitation rates or dry antecedent conditions). Small variations in the coefficients of a strong signal can sometimes be offset by larger variations in the coefficients of a weak signal, and least-squares fitting will make such a tradeoff if it leads to a closer match to the observed streamflow.  ERRA does its best to suppress statistical cross-talk between the coefficients of the strong signals and the weak ones, but this is an inherently difficult task.  Because streamflow is relatively insensitive to low-intensity precipitation and rain that falls under dry conditions, it is difficult to estimate the corresponding coefficients accurately, particularly at long lags.  Nevertheless, Fig. 12 clearly shows the main peak of the runoff response to both high-intensity and low-intensity precipitation falling under both wet and dry ambient conditions." (lines 704-722)